# Northward drift of the Azores plume in the Earth's mantle

Maëlis Arnould [1,2,3], Jérôme Ganne[4], Nicolas Coltice[1] & Xiaojun Feng [5]

Mantle plume fixity has long been a cornerstone assumption to reconstruct past tectonic plate motions. However, precise geochronological and paleomagnetic data along Pacific continuous hotspot tracks have revealed substantial drift of the Hawaiian plume. The question remains for evidence of drift for other mantle plumes. Here, we use plume-derived basalts from the Mid-Atlantic ridge to confirm that the upper-mantle thermal anomaly associated with the Azores plume is asymmetric, spreading over ~2,000 km southwards and ~600 km northwards. Using for the first time a 3D-spherical mantle convection where plumes, ridges and plates interact in a fully dynamic way, we suggest that the extent, shape and asymmetry of this anomaly is a consequence of the Azores plume moving northwards by 1–2 cm/yr during the past 85 Ma, independently from other Atlantic plumes. Our findings suggest redefining the Azores hotspot track and open the way for identifying how plumes drift within the mantle.

[1] Laboratoire de Géologie, École Normale Supérieure, CNRS UMR 8538, PSL Research University, 75005 Paris, France. [2] Laboratoire de Géologie de Lyon, Terre, Planètes, Environnement, École Normale Supérieure de Lyon, Université de Lyon, Université Claude Bernard, CNRS UMR 5276, 2 rue Raphaël Dubois, 69622 Villeurbanne, France. [3] EarthByte Group, School of Geosciences, Madsen Building F09, University of Sydney, Sydney 2006 NSW, Australia. [4] IRD, CNRS, GET, Université Toulouse III, 14 Avenue Edouard Belin, 31400 Toulouse, France. [5] School of Safety Engineering, China University of Mining and Technology, Jiangsu 221116, China. Correspondence and requests for materials should be addressed to M.A. (email: arnould@geologie.ens.fr)

Although fixed hotspots[1] have long been considered as a reference for plate reconstructions[2], paleomagnetic record and age progression along hotspot tracks have suggested substantial drift of Pacific plumes, questioning our conception of static hotspots. Nevertheless, identifying the individual motion of a plume usually requires a long and continuous chain of seamounts, and precise geochronological and paleomagnetic data, which explains that Hawaii is the only plume with clear evidence of drift so far[3,4]. We propose in the following that defining the shape of upper-mantle temperature anomalies provides evidence for plume drift. We take the case of the Azores mantle plume as an ideal case.

The Azores plateau is an oceanic area of about 150,000 km[25], with a magmatic center located about 100–200 km[6,7] east from the Mid-Atlantic Ridge (MAR) axis (Fig. 1a). The spatial correlation between bathymetric highs[7,8], anomalously thick crust[9,10], gravity anomalies[11], and geochemical signatures[12] along the MAR (Fig. 2c–e) point to an interaction between the ridge and a deep mantle plume. Both upper-mantle[13–15] and global seismic tomography studies[16] describe the presence of a rather vertical low seismic velocity conduit beneath the Azores. This seismic anomaly extends in the lower mantle[16] and the primitive $^4$He/$^3$He ratios of the lavas[17] further consolidate the deep origin of the Azores plume.

Symptoms of the plume-ridge interaction spread over a large region. The bathymetric high along the ridge forms an asymmetric bulge declining by 2.5 km over 500 km toward the north, and by 3 km over 2000 km toward the south[7,8], consistent with Bouguer anomalies[11] and plume geochemical fingerprints (key trace element ratios)[12], (Fig. 2c–e). V-shaped gravity and topographic structures were detected only south of the Azores[18], supporting the presence of a southward-elongated asymmetry. High resolution global[19] (Fig. 1b–d) and regional[15] tomographic models of the upper mantle show the asymmetric slow velocity region extends down to at least 150 km depth and spreads along the ridge over 3000 km. The estimates for the Azores plume volume flux vary between 51 m$^3$/s, based on the spreading rate of the MAR[20] to 90 m$^3$/s, deduced from the topographic swell around the hotspot[21]. Theories for interactions between a ridge and a static plume[22–24] fail to explain the extent of hot plume material spreading along the ridge (called waist) and its volume flux: accounting for a waist >2000 km consistent with bathymetry[7], gravity[11], geochemistry[12] (Fig. 2c) and seismic tomography[19] (Fig. 1) would lead to a volume flux of about 400 m$^3$/s[25], at least five times greater than current estimates for the Azores plume[20,26]. The presence of anomalously buoyant plume material more than 2000 km south of the plume cannot either be explained by the burst of plume activity 10 Ma ago[18], suggested by the existence of V-shaped anomalies south of the Azores. This would only have led to a local increase of $T_P$ of about 70 °C, at about 600 km from the hotspot location and lasting only a few million years[24].

Here we present first potential temperature ($T_P$) estimates along the MAR using the PRIMELT method that confirms the extent of the thermal waist of the Azores plume. We then investigate the dynamic connection between the plume asymmetry and relative plate-plume motions with 3D-spherical models of convection with plate-like behavior at Earth's-like convective vigor. Accounting for the kinematic context around the Azores, our model suggests that a northward motion of the plume >1 cm/yr explains the geometry of the observed asymmetry.

## Results
### New evidence for a thermal asymmetry below the Azores.
MORBs are extracted from a partially melted region in the shallow ambient convective mantle. Their chemistry likely reflects the potential temperature ($T_P$) of this region, corresponding to the temperature that the ambient mantle would reach, should it raise adiabatically to the Earth surface without melting[25]. Global systematics of the chemistry of MORBs suggest $T_P$ variations of 200–250 °C along, and immediately below, mid-oceanic ridges[9,14], challenging some previous estimates (~100 °C[27,28]). However, whether or not these variations simply reflect plume-ridge interactions, chemical heterogeneities in the mantle source, inappropriate corrections for fractionation[29] or the presence of fluids in the magmatic source[30] remains a source of debate.

A more selective but secure approach relies on the chemical composition of primary magma formed by melting of dry, fertile peridotite, with limited fractional crystallization prior to magma emplacement in the crust, using the PRIMELT method[31,32]. Inferences, though limited, already suggest that MORBs often appear too differentiated to yield $T_P$ estimates with the PRIMELT method. Conversely, intraplate volcanism, which also gives birth to primary magmas through decompression melting of deeper and hotter sections of the mantle (e.g. mantle plume), can provide successful solutions of calculation with PRIMELT. It is therefore a powerful tool to study the extent of plume thermal imprint along ridges.

Geochemical data considered in this study come from Gale et al.[33], who compiled an extensive chemical database of near zero-age (i.e. <ca. 300 kyr) MORBs collected along the axis of mid-oceanic ridges (Supplementary Figs. 1 to 7 and Supplementary Note 1). We focus on the northern sections of the MAR (Fig. 2), from latitude 0 to 90°N where data is the most abundant. Figure 2b illustrates the distribution of $T_P$ obtained with PRIMELT along the MAR in the Azores area. A temperature anomaly spreads over 2000 km southwards the Azores hotspot while it disappears about 600 km northwards of the hotspot. Figure 2a, b and Supplementary Fig. 8 show that the temperature decrease correlates with a chemical depletion in calcium in the studied MORBs. Calcium depletion is indicative of a colder lithospheric or asthenospheric mantle below the MAR since it can result either from increasing melting of pyroxenite in the mantle favored by its lower solidus relative to peridotites or from increasing fractionation of magmatic pyroxenes during the ascent of primary melts through the lithospheric column, their crystallization being primarily controlled by the temperature of the mantle or the crustal section crossed by the melts. Petrological equilibrium curves[34] show that MORBs sampled away from the Azores crystallized pyroxenes at greater pressures and that the decrease in pressure correlates with the distance to the hotspot (Fig. 2a). This supports the idea of a temperature control on calcium depletion in MORBs and suggests that beneath the Azores, pyroxenes crystallize mainly in the oceanic crust due to the presence of anomalously hot mantle while further away, pyroxenes mostly crystallize in the mantle section (Supplementary Figs. 4 and 5).

### Thermal asymmetry caused by plume-ridge relative motions.
We explore the hypothesis that the asymmetric thermal anomaly below the MAR corresponds to the thermal wake of the Azores plume, formed by shearing caused by the relative motion between the plate system around the ridge and the plume[35]. To do so, we analyze the uppermost mantle thermal signature of drifting mantle plumes naturally emerging from a 3D-spherical convection model with self-consistent plate-tectonics and interacting with ridges (see Methods). In such models, plumes interact in a fully dynamic fashion with surface tectonics and convection currents at all scales[36]. Hence, our model also takes into account in a fully dynamic way the effects of plume-ridge interactions described in previous regional models[21–23]. Instead of using

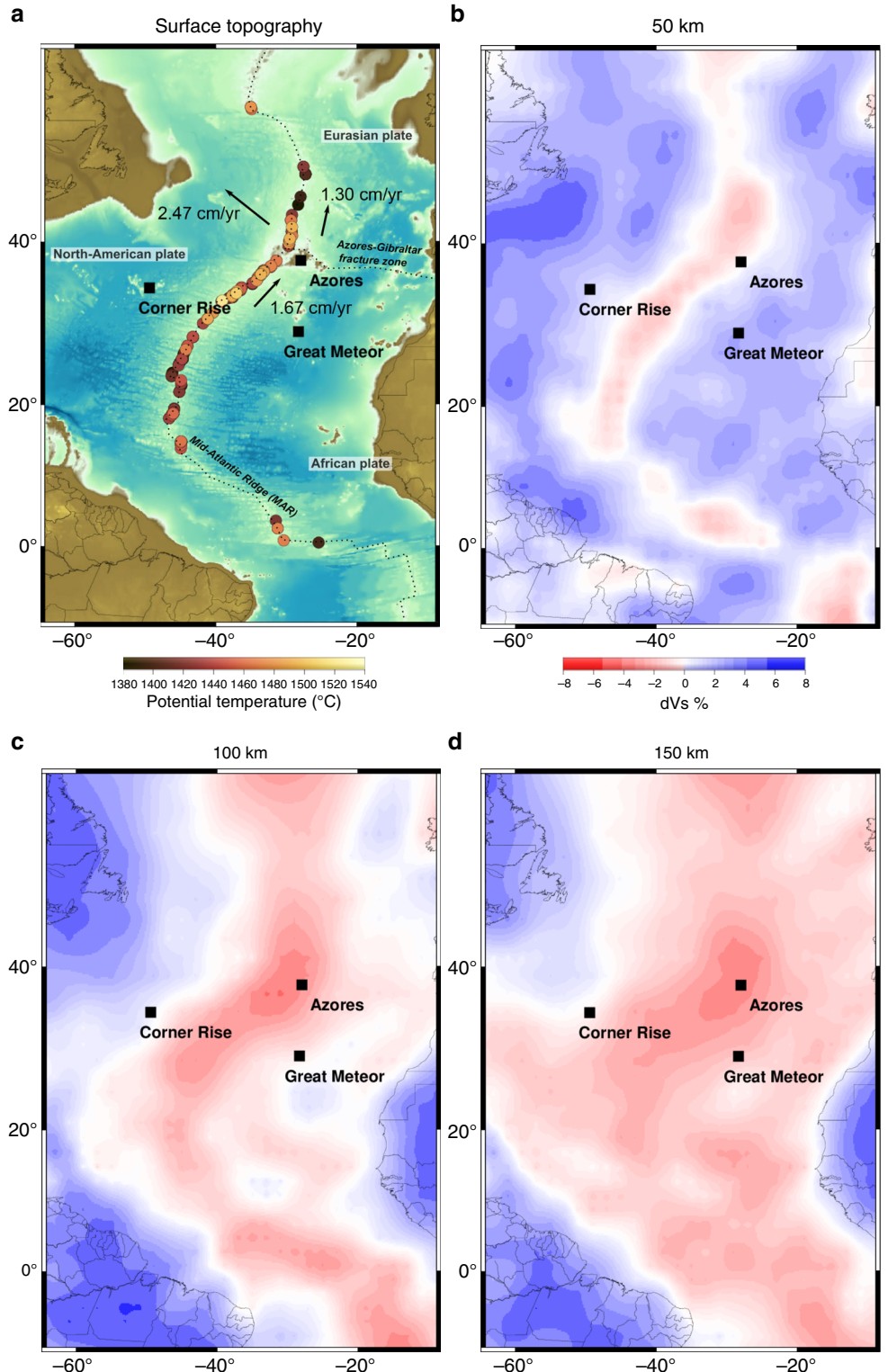

**Fig. 1** Geographical setting and seismic tomography of the upper mantle. **a** Surface topography of the Central Atlantic Ocean and potential temperatures obtained from the PRIMELT method along the MAR (see Fig. 2). The direction and magnitude (in cm/yr) of plate velocities according to their absolute motions during the last 10 Ma in a moving hotspot reference frame[42] in the vicinity of the Azores are denoted by the black arrows and numbers. **b**–**d** Upper mantle shear wave tomography slices from the global upper mantle and transition zone model 3D2017_09SV[19]

multiple regional models and imposing potentially unphysical constraints to our system to generate relative motions below a ridge, we are able to study a large number of self-evolving interactions between ridges and plumes by computing a global numerical model producing surface velocities, heat flow

and plate size distributions comparable to Earth. We detect about 25 plumes on average at each time step. These plumes have a $210 \pm 45\,°C$ average temperature excess with respect to the ambient mantle, an upper mantle vertical velocity of $41 \pm 20$ cm/yr, conduit diameters of $200 \pm 150$ km, and volume

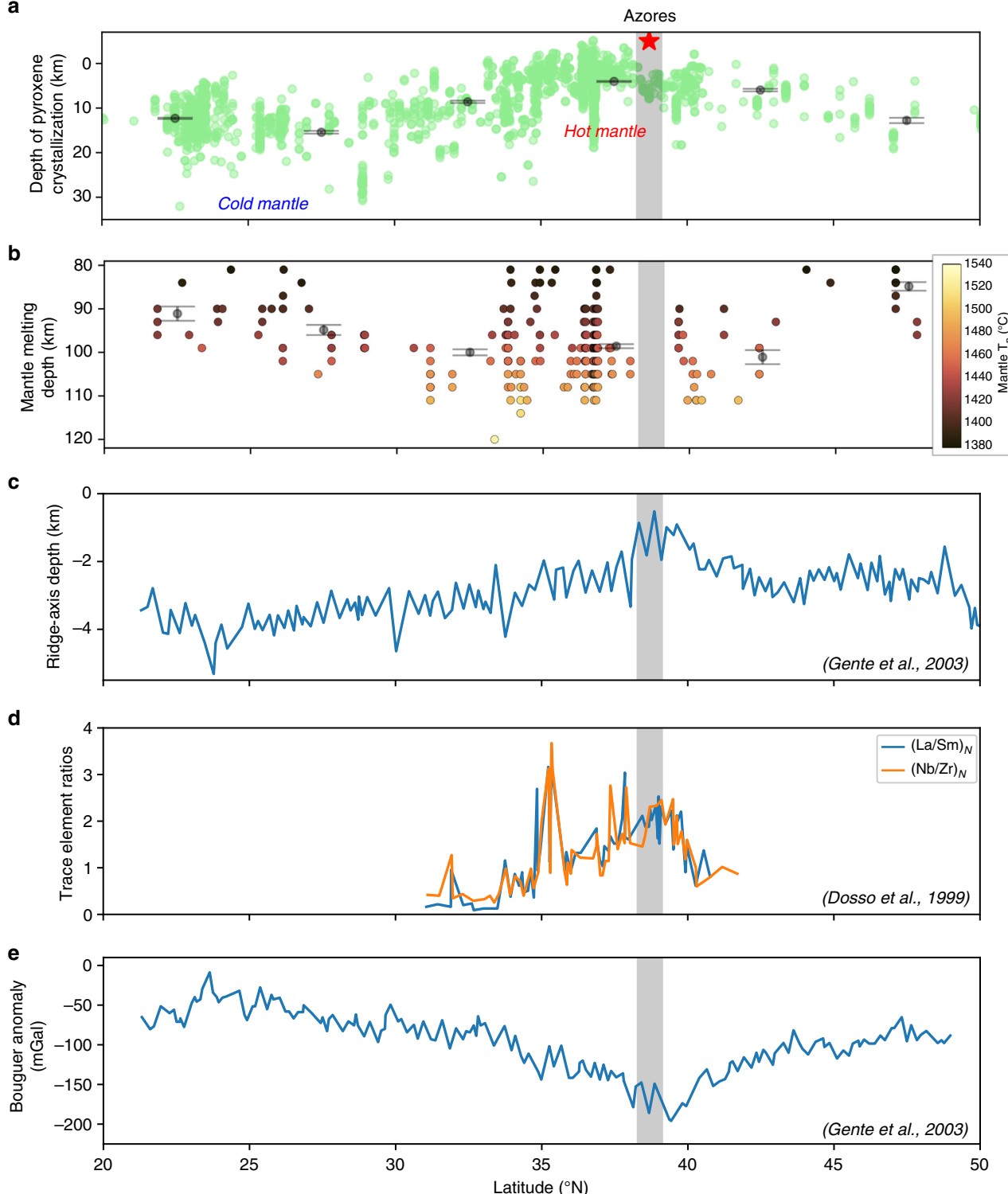

**Fig. 2** Chemical, thermal, bathymetric, geochemical and gravity asymmetry along the Mid-Atlantic Ridge. **a** Depth of pyroxene crystallization then subtraction (e.g. fractionation) in a primary magma that, later, will give birth to MORBs, calculated using Equation (6) in Herzberg[34] and assuming a ratio of 1:3 between pressure (kb) and depth (km). **b** Mantle depth at which adiabatically upwelling mantle below the MAR segments crossed their solidus, depending on their potential temperature ($T_P$) calculated with PRIMELT3 MEGA software[60] using reduced conditions ($Fe^{2+}/\sum Fe = 0.9$) in the source. Solutions of calculation have been filtered for MgO < 8 wt% (blue dots in Supplementary Fig. 2). Mean values and the associated standard deviations (grey points and bars) in **a**, **b** are obtained by bootstrap analysis, are reported at 5° step of latitude. **c** Ridge-axis depth (below sea-level) in the region of the Azores[45]. **d** Geochemical trace element ratios along the MAR[12], normalized according to Bougault and Treuil[61]. **e** Ridge-axis Bouguer anomaly[45]. Data have been plotted against latitude along the x-axis

fluxes of $40 \pm 15$ m$^3$/s (Supplementary Fig. 9). We identify a total of 76 distinct plumes over the 320 Myr of model duration, moving laterally relative to a no-net-rotation surface at $1.75 \pm 0.38$ cm/yr on average. Relative motions between hotspots are limited to $1.2 \pm 0.4$ cm/yr. We observe that 46 of them interact at some point with ridges, either ponding below them or contributing to their propagation.

In our models, the relative motion between a considered plume and plates around the ridge shears the plume and generates thermal asymmetry. We identify four kinematic cases. In case 1 (Fig. 3a), the plume moves along the ridge in the opposite direction to the average direction of plate motion. Hence shearing always occurs in the direction of plate motion, building up a highly asymmetric wake. In case 2, both the plates and the plume move in the same direction. Shearing depends on which of the plates or the plume moves faster. If the plume does, like in Fig. 3b, shearing is opposite to the direction of the plates. If the plates do, shearing is in the direction of plate motion. If both move at about the same velocity, there is no asymmetry. In case 3 (Fig. 3c.), the plume moves perpendicularly to the ridge. Shearing along the ridge is then negligible and there is no asymmetry in this direction, although it can exist in the direction of seafloor spreading. Finally, in case 4 (Fig. 3d), the plume lies below a ridge close to a ridge–ridge–ridge triple junction. The dominating effect causing thermal asymmetry is the channelization of the wake material toward the triple junction.

Figure 3f and Supplementary Fig. 10 show that temporal changes in the relative motion of the plate system and a plume can lead to a reversal of the asymmetry over a timescale of 50 Myr.

## Discussion

In the case of the Azores, the asymmetry of the thermal anomaly extends southwards. The south–westward orientation of the MAR south the Azores cannot explain by itself the much larger extent of the thermal anomaly of the plume to the south (Fig. 1a). Therefore, explaining the asymmetry of the Azores plume would either require that the African and the North-American plates move southwards relatively to the Azores (case 1, Fig. 3a), or that both the Azores plume and those plates move northward, the plume moving at a faster pace (case 2, Fig. 3b). Therefore, we compared the observed thermal asymmetry with the motion of the African, European, and North-American plates in several reference frames based on moving hotspots. A southward motion of Africa was proposed by the NUVEL 1A-HS3 reference frame[37], based on Pacific hotspot tracks. However, this reference frame does not constrain the tracks of South-Atlantic hotspots. Using a global moving hotspot reference frame, Steinberger et al.[38], Torsvik et al.[39] and Doubrovine et al.[40] proposed stable African and European plates and suggested a westward-drifting North-America by 2 cm/yr (corresponding to case 3, Fig. 3c). Accounting for a stationary or eastward-moving plume would generate a westward-oriented asymmetric wake which would result in a symmetric thermal anomaly along the MAR axis (Fig. 3e). Therefore, only a combination between a northward motion of the Azores plume and such plate motions would explain the observed southward asymmetry of its wake. In the moving hotspot reference accounting for Indo-Atlantic hotspot tracks, O'Neill[41] and Torsvik et al.[42] reconstructed a northward motion of Africa at a speed larger than 1 cm/yr in the vicinity of the Azores (Fig. 1a). In that case again, the Azores situation compares to modeled case 2 (Fig. 3b): the asymmetry results from a northward motion of the plume, at a faster rate than the plates (>1 cm/yr). Finally, the asymmetric motions of Africa and North-America predict a westward migration of the MAR axis during

the last 50 Ma at a speed of about 1.5 cm/yr[6]. Although this would possibly result in a shift between the location of the wake of the Azores plume and the MAR, our model shows that the potential temperature along a ridge having interacted with a plume at least 40 Myr before still records the asymmetry related to the relative motion between the ridge and the plume (Supplementary Fig. 11). The northward motion of the Azores plume appears at odds with South-Atlantic hotspots (Supplementary Fig. 12), which are commonly viewed as stable, based on the geometry of their hotspot tracks[43]. This result differs from previous estimates of the Azores drift from numerical models of instantaneous flow induced by a density distribution based on seismic tomography[44], which favor a southwestward motion of the plume. Although such models have matched the actual position of plumes on Earth, the predicted flow close to the core-mantle boundary where the plume moves is uncertain. Indeed, in the deepest mantle, tomographic models have relatively low resolution, show significant differences, and the conversion of seismic anomalies to density is ambiguous. Moreover, the computed flows do not consider lateral viscosity variations, which are dominant in this area where cold slabs and hot plumes potentially interact. Finally, the ascent of plumes is not computed self-consistently in the flow model contrarily to our models. These models fail to explain the southward-elongated asymmetry across the Azores since they would predict an asymmetry in the opposite direction.

Since the Azores wake is highly asymmetric (Figs. 2 and 3f), it also suggests that the direction of the relative motion between the plume and the plates has remained stable over a timescale longer than a few tens of Myr. A long-term northward motion of the Azores plume relative to the plate system is consistent with the proposed scenario of the Azores plume building up the Corner Rise and Great Meteor plateaus[45] (located about 1000 km away from the present-day location of the plume, Fig. 1a). The direct K–Ar[46] and indirect[47] dating of these complexes were traditionally used to affiliate them to the New-England hotspot track mainly located on the North-American plate[2], despite major uncertainties on the age determination of these seamounts (ranging from 11 to 65 Ma for Great Meteor). However, recent tectonic[45], geochronological and geochemical[48] studies bring complementary information for a genetic link with the Azores. A northward motion of the Azores plume at a rate of 1–2 cm/yr is consistent with such scenario of building a track from 85 to 20 Ma and the proposed migration of the plume away from the MAR at $10 \pm 5$ Ma[45].

Our study shows that characterizing thermal anomalies linked with plume-ridge interactions has the potential to unravel absolute plume motions, questioning our knowledge of mantle plumes and their thermal imprint in the upper mantle and along ridge segments[14,24,31]. The proposed motion of the Azores plume is consistent with other evidence of moving mantle plumes[4,49]. Nevertheless, the causes for such motions are still unclear. The possible anchorage of the Azores plume on the edge of a dense and stable large low shear velocity province (LLSVP), as suggested in tomographic models[50], is not accounted by our convection model. However, convection calculations do not show enhanced fixity of plumes with chemical anomalies at the core-mantle boundary[51]. Also, slow deformation of the edges of LLSVPs produce plume motions of the order of 1 cm/yr[52], and plumes can propagate longitudinally along the edges of LLSVPs which undergo significant thermomechanical or thermochemical instabilities[53].

## Method

**Primelt**. Mantle $T_P$ calculation using PRIMELT is based on the MgO content of the primary magma, and MgO typically correlates with FeO. The calculations require that FeO be obtained from FeO$_T$ (total iron reported as a single oxide)

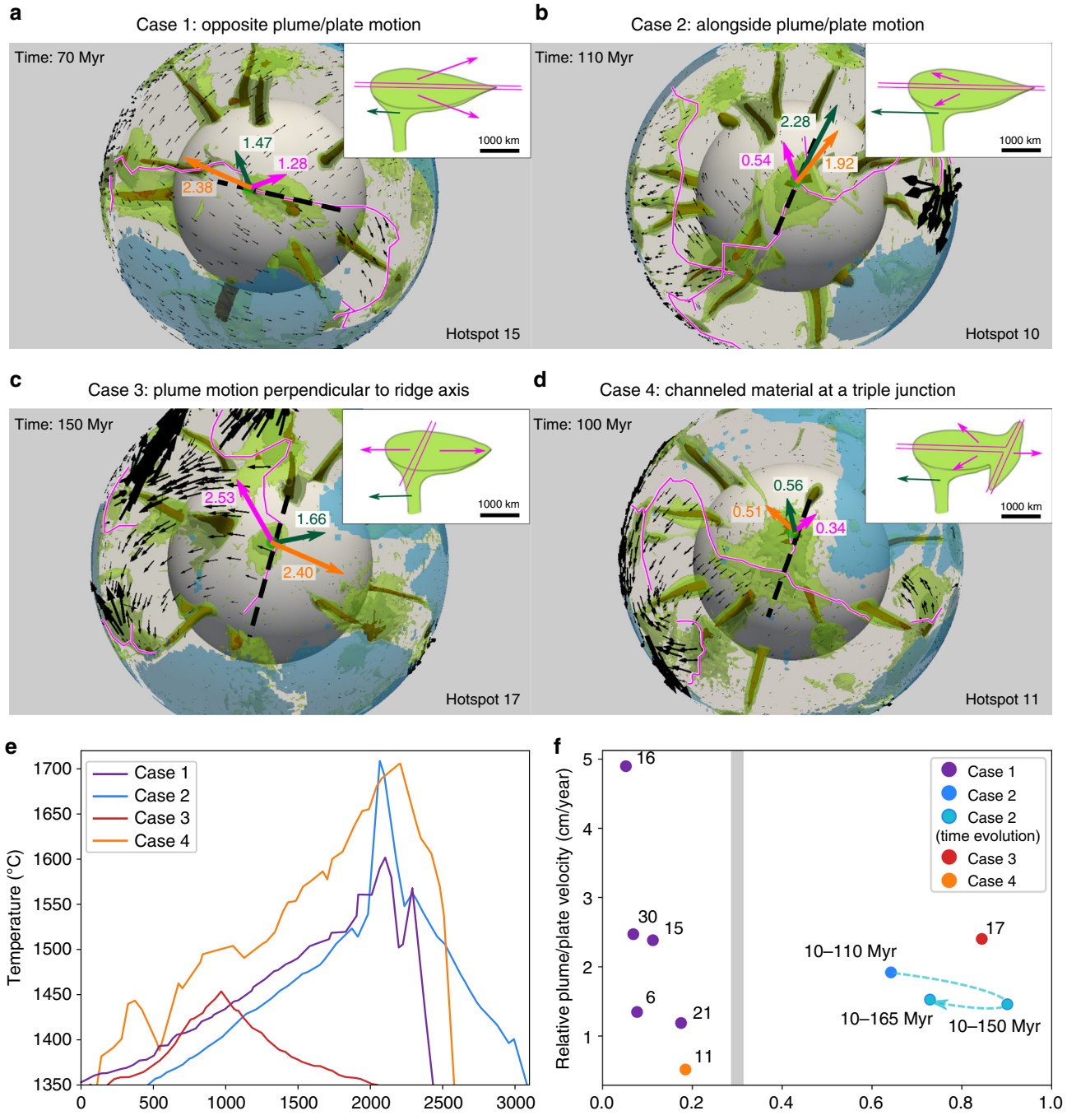

**Fig. 3** Modeled kinematic scenarii of the interaction between a plume and a ridge system. **a–d** Snapshots of four modeled cases of the relative motion between a ridge system (magenta lines) and a mantle plume (outlined by the light green (1390 °C), dark green (1480 °C), and brown (1700 °C) isotherms) in a global mantle convection model self-generating plate-like tectonics. Transparent green circles show the location of the plume thermal maximum. Colored arrows centered on the plume with magnitudes in cm/yr correspond to the plume absolute velocity (dark green), the average plates velocity within 500 km of the plume center (magenta) and the relative velocity between the plume and the average plates velocity (orange). Black arrows scale with surface plate velocities and blue transparent areas show the location of continents. Insets are schematic diagrams of the corresponding kinematic cases. **e** Temperature profiles across the plume waist at 100 km depth and along the modeled ridge (black dashed lines in **a–d**). **f** Magnitude of the relative velocity between plumes and plate as a function of the symmetry of the temperature profiles of corresponding plumes (Supplementary Fig. 14). Symmetry coefficients correspond to the ratio of the extent of the thermal anomaly on each side of the corresponding plume thermal maximum. Numbers correspond to plume IDs. The vertical grey bar corresponds to the asymmetry observed at the Azores. Source data are available in the Open Science Framework repository[62]

and this can be estimated assuming $Fe_2O_3/TiO_2 = 0.5$ or assuming a constant redox condition in the magmatic source ($Fe^{2+}/\Sigma Fe = 0.9$ or $0.8$). We tested different hypotheses. Results are given in Supplementary Figure 1. Uncertainties that arise by calculating FeO using $Fe_2O_3/TiO_2 = 0.5$ instead of $FeO/FeO_T = 0.9$ propagate to uncertainties in mantle $T_P$ lower than 30 °C for high-Ti type lavas ($TiO_2 > 1$ wt%) up to 60 °C for low-Ti types ($TiO_2 < 0.5$ wt%). MORBs are

dominantly characterized by Ti content >1 wt% (Supplementary Fig. 1d). As such, alternative adoption of a Ti-dependent redox condition is unlikely to affect our calculation toward more elevated $T_P$. Consistently, a more oxidizing ratio of $Fe^{2+}/\Sigma Fe = 0.8$ will decrease FeO in the primary magma, yielding lower MgO and lower mantle $T_P$ but such general decrease will not change our general conclusions because it involves the whole set of data.

The petrological software is calibrated by anhydrous melting experiments on fertile peridotite, and its application to lavas assumes a similar fertile and dry peridotite source. Uncertainties in fertile peridotite composition do not lead to significant errors in mantle $T_P$[54]. Conversely, melting of wet peridotite is likely to result in MgO values that are too elevated. We excluded all samples that have undergone augite and/or plagioclase fractionation as indicated by depletion of CaO and $Al_2O_3$ or enrichment in FeO. Such discrepancies can be highlighted in a simple graph by plotting the chemistry of sample against its "liquid line of descent" (i.e. LLD, Supplementary Fig. 2c). Our calculations have been filtered according to a graphic procedure[54] leading to reduced uncertainties that arise from pyroxenite source lithology, source volatile content from metasomatized peridotite, and clinopyroxene fractionation (Supplementary Fig. 3). We assume that the remaining samples (~11%) have been affected only by minor amounts of olivine addition and subtraction. Solutions of our calculation are provided in the supplementary material (Supplementary Data 1).

**Numerical modeling**. We use the thermochemical convection code StagYY[55] to model mantle convection in 3D-spherical geometry with continental lithospheric rafts and self-generation of plate-like tectonics at its surface. We solve the equations of mass, momentum and energy conservation under the Boussinesq approximation:

$$\nabla.\mathbf{v} = 0$$

$$\nabla p - \nabla.\left[\eta\left(\underline{\nabla\mathbf{v}} + (\underline{\nabla\mathbf{v}})^T\right)\right] = \mathrm{Ra}(\alpha(z)T + BC)\mathbf{e_r}$$

$$\frac{DT}{Dt} = -\nabla.(\nabla T) + H$$

$$\frac{DC}{Dt} = 0$$

where $\mathbf{v}$ is velocity, $p$ is pressure, $\eta$ is viscosity, $T$ is temperature and $C$ is composition. $H$ is the mantle internal heat production rate, $\alpha$ is a depth-dependent thermal expansivity coefficient set as in Arnould et al. 2018[56], $B$ is the chemical buoyancy ratio and $\mathbf{e_r}$ is the radial unit vector. The Rayleigh number of the system is:

$$\mathrm{Ra} = \frac{\alpha_0 D^3 \Delta T \rho_0 g}{\kappa \eta_0}$$

where $\alpha_0$ is the surface thermal expansivity, $D$ the mantle's thickness, $g$ the gravitational acceleration, $\Delta T$ the temperature gradient across the mantle, $\rho_0$ the reference density, $\eta_0$ the reference viscosity and $\kappa$ the reference thermal diffusivity. In the studied model, Ra is $10^7$, which results in a convective vigor comparable to the Earth. All mantle parameters are constant with depth, except the thermal expansivity coefficient (decreasing by a factor of 3 throughout the mantle), and the viscosity, which varies with temperature $T$ and pressure $P$ as:

$$\eta(z, T) = \eta_0(z).\exp\left(A + \frac{E_a + \Pi(z)V_a}{RT}\right)$$

where $A$ is a constant so that $\eta = \eta_0$ for $T = 0.64$ (corresponding to 1600 K), which is the a priori temperature at the lithosphere-asthenosphere boundary, $\Pi(z)$ the static pressure, $R$ the gas constant and $T$ the temperature. The activation energy is 146 kJ/mol, implying 7 orders of viscosity variations throughout the domain. The activation volume is 13.8 cm³/mol, and we impose a viscosity jump by a factor of 30 at 660 km, consistently with postglacial rebound and geoid modeling[57,58]. Average temperature and viscosity profiles are shown in Supplementary Fig. 13.

We introduce pseudo-plasticity[59] to generate plate-like behaviour (plate boundaries emerge, they are not imposed). After exploration of the parameter space, we choose a yield stress value of 48 MPa for the oceanic lithosphere, so that the dynamic regime produces a plate-like behavior. The surface yield stress is 10 and 20 times higher for the margins and the interior of continental rafts respectively, to avoid their excessive mechanical erosion. The yield stress varies with depth by a factor of 1.8 MPa/km for all materials. The parameters are described in Supplementary Table 1.

We first run the model to reach a thermal statistical steady state. Then, we compute a total evolution of 320 Myr. Such model is at the limit of computation capabilities and required about 7 weeks of computing time on a parallel supercomputer of the National Computational Infrastructure (Australia). In the numerical solution, mantle plumes originate from the CMB where the heat flow is about 5 TW. Internal heat is produced at a rate of about $8.5 \times 10^{-12}$ W/kg. Surface heat flow reaches 34 TW, and the root-mean square surface velocity is 3.26 cm/yr. Plumes and plate boundaries are emergent structures, not imposed in such models. All plumes can spontaneously move relatively to a no-net rotating surface. After the run, we examine these structures produced in a self-consistent manner by the system of equations.

## Data availability

Source data for Figs. 1a, 2a, b are available in Supplementary Data 1. Source data for Fig. 3 are available in the Open Science Framework repository (DOI 10.17605/OSF.IO/93MEQ).

## Code availability

The convection code StagYY is the property of Paul Tackley. and Eidgenössische Technische Hochschule (ETH) Zürich. It is available on request from Paul Tackley. (paul.tackley@erdw.ethz.ch).

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

## Acknowledgements

The research leading to these results has received funding from the IRD, CNRS-INSU, and the European Research Council within the framework of the SP2-Ideas Program ERC-2013-CoG, under ERC grant agreement 617588. Calculations were performed with the assistance of resources from the National Computational Infrastructure (NCI) through the National Computational Merit Allocation Scheme supported by the Australian Government. The perceptually uniform colour map lajolla from Fabio Crameri is used in Figs. 1a and 2b to prevent visual distortion of the data.

## Author contributions

M.A. and J.G. conceived the study and wrote the paper. N.C. helped to generate the research idea and contributed to the writing and focusing of the paper. J.G. and X.F. contributed to the petrologic analysis and PRIMELT modelling. M.A and N.C. developed the numerical models and the kinematic analysis of the Azores plume motion. All authors contributed to the interpretation of results.

## Additional information

**Competing interests:** The authors declare no competing interests.

