## [Peer Review File · Nature Communications]

Reviewers' comments:

Reviewer #2 (Remarks to the Author):

This manuscript provides a novel means to support a general northward drift of the Azores plume since ~85 Ma. The conclusions are based on defining an asymmetric mantle potential temperature profile across the northern Mid-Atlantic ridge then comparing the results to model generated plume-ridge interactions. This provides an interesting insights into the Azores plume as well as provides a new method of studying plume-ridge interaction which will be of interest mantle geo-chemists, dynamicists and physicists. This manuscript is suitable for publication in Nature Communication with some revisions.

Major Points

Please segment the manuscript into more discrete sections (e.g. Introduction, Results and Discussion, Conclusion) to improve readability.

The authors argue the 1-2 cm/yr drift of the plume is consistent with the Corner Seamounts being sourced by the Azores plume. However, these seamounts are not dated (to my knowledge) and the ages are modeled based on assuming the seamounts belong to the New England Seamount Chain (Duncan 1984) or backtracked based on sea level erosion (Tucholke and Smoot, 1990). This assumption should be made clear in the manuscript as significant uncertainty remains over the ages of those seamounts. Further discussion of the plate motion – plume motion combination needed to link these Seamounts to the Azores plume is required.

Figures need title sentences.

Supplementary figures 3 through 7 and the corresponding impact on the results/discussion use “25% of the dataset”. The authors need to describe what 25% of the data they used and how they filtered/selected the data. Were outlier removed? Was the data filtered at random? Was the entire dataset used in the calculations but only 25% shown in the figures? This has the potential (although I doubt it will change the results) to influence the mantle potential temperature estimates depending on where data was omitted.

I feel some mention of the difference in Azores plume motion obtained by your methods (northward) and by mantle flow modeling based on mantle tomography models (Steinberger 2000-JGR) is warranted. Steinberger modeled a Southeast ward drift of the plume since 80 Ma. Although there are uncertainties and assumptions involved in Steinberger (2000)'s method I feel your Northward drift results warrant some comparison.

On lines 129-132 the authors argue that the observed results from the modeled Ridge axis 2 compares to the modern day results for the Azores ridge. An addition to figure 4 or new figure 5 comparing the model results of potential mantle temperature for some distance across the ridge directly to the observed data would strengthen this argument. This may not be feasible so I don't require this addition.

Line by line points:

Line 36: Azores plume contains low $4\text{He}/3\text{He}$ (or high $3\text{He}/4\text{He}$) anomalies as evidence for a more primitive mantle source. The high $4\text{He}/3\text{He}$ lava flows (more radiogenic) do not necessarily indicate a deep mantle source.

Lines 81 – 83: Seems like a large portion of the data indicate that beneath Azores and Iceland, pyroxene crystallizes in the mantle as well as the crust. Please elaborate or provide appropriate references on how “pyroxenes crystallize only in the oceanic crust”.

Lines 88 – 95: Awkward structure/wording.

Lines 178-202: This read as additional discussion more than methods. I recommend moving it to the start of the supplemental document.

Supp figure 5: I'm confused by the x-axis. I think it should be degree West (?) as Iceland sits at ~19°W. More clarity on what the x-axis represents is required. Do all the MORB samples > ~19° East contain MgO contents < 8 wt% or was depth of melting just not calculated for those samples? Please clarify in figure caption.

Supp figure 6: Where does the crustal age data come from? Reference required.

Pedantic/Minor points:

Inconsistent comma use for integers over 1000 (e.g. line 40 says '3000 km', line 42 uses '3,000 km')

Sup Figure 2: Potential is capitalized in the second line.

Supp figure 5: Third line, there are two parentheses before 'here'.

Supp figure 5: Below is misspelled as 'Bellow'

Supp figure 6: Here is misspelled as 'Hhere'

Figure 3: Crystallization is misspelled on the y-axis of (a).

Line 157: Awkward wording for the "We tested different..." sentence.

Line 115: Space needed between '0.38' and 'cm/y'

Thanks for the interesting read,
Kevin Konrad

Reviewer #3 (Remarks to the Author):

This paper supposedly shows evidence for northward drift of the Azores plume in the Earth's mantle. However, I find the argumentation unconvincing and therefore cannot recommend publication. Their main argument is that there is a thermal anomaly associated with the Azores hotspot, and this anomaly stretches further southward than northward. Their conclusion is that the plume must have been further south in the past to cause this thermal anomaly there. This sounds convincing at first sight, and in fact plume-related temperature and lithospheric heat flow anomalies have been proposed elsewhere, but after thinking more carefully, I concluded that it actually cannot be true. This is, because the lithosphere at the ridge didn't exist 85 Myr ago. With a half-spreading rate of the order of 1 cm/yr, the lithosphere that was at the ridge 85 Myr ago is now about 850 km away. Likewise, because of mass conservation, material that is now at the ridge (and causing the higher temperatures) would have been a similar distance away from the ridge back then. So the thermal structure that is there now has likely nothing to do with the location of the plume in the distance past.

Also, I don't find the numerical experiment convincing to support that case. Supposedly (line 118) the plume moves with 1 cm/yr over 70 My, which would be about 700 km, or 600 km in 60 Myr. However, in Figure 4a I measured that the center of the plume is 2.3 cm from the left margin, and 1.4 cm from the bottom margin. Within my measurement uncertainty, I found it is at the same location also at 40 Myr and at 70 My. For the diameter of the Earth I measured 3.8 cm, corresponding to 12742 km. So 1 mm corresponds to 335 km, and a 600 km motion should correspond to nearly 2 mm difference and definitely be visible, but it isn't. Only at 100 Myr, I find the plume slightly ENE at 2.5 cm from the left and 1.45 cm from the bottom margin. But the asymmetry of the temperature anomaly is already there at 10, 40 and

70 Myr, so it cannot have been caused by motion of the plume. I find it much more likely that the asymmetry is caused by underlying mantle flow that drags plume material preferentially in one direction. In fact the plate motions shown would indicate just that to me: The plate to the west shows strong westward motion, and this would presumably correspond to underlying mantle flow in a similar direction. Similarly, I would argue for the same mechanism working for the Azores hotspot: The closest extensive subduction zones over the past few tens of Myr, corresponding to extensive slabs as seen through tomography, are to the southwest. Hence those sinking slabs would drive southwestward large-scale flow in the direction along the ridge. And in fact geodynamic models based on tomography show just that. So you cannot write in the abstract "we show that the Azores plume moved northward by 1-2 cm/yr from 85 Myr", because this is just your interpretation of results, but I am sure it is wrong.

Also, I find the interpretation of observational data unconvincing. Not that I would understand any of this (as I'm not a geochemist) but for example you conclude from Figure 3b a decrease by $0.03^{\circ}\text{C}/\text{km}$ towards the south and $0.12^{\circ}\text{C}/\text{km}$ (on line 70) or even $0.8^{\circ}\text{C}/\text{km}$ (on line 131) towards the north. But as far as I can see, this is just eyeballing, and no proper statistics is being done.

Also, they interpret the Corner rise and Great Meteor oceanic plateaus as part of the Azores hotspot track. However, the usual interpretation (e.g. Tucholke and Smoot) is that the Great Meteor Seamount is at the young end of a hotspot track that also includes seamounts to the north of it. This is also supported by relatively young radiometric ages from Great Meteor seamount, indicating that it was formed ~ 20 Myr ago. Also, this trend fits e.g. the plate motion shown in Figure 8 SI. So I find the "traditional" interpretation -- that Great Meteor is a separate hotspot (perhaps now extinct) that first created a track on the North American plate (starting in New England, followed by New England and Corner Seamounts, then was overridden by the ridge, and subsequently created a track on the African plate, with its young end to the South) much more convincing and compatible with observations. Also, they argue that there is evidence that the Hawaii hotspot has moved, and likewise, the Azores hotspot should also have moved. However, one can find perfectly good reasons why the Hawaii hotspot should have moved much more: Because it is beneath the fast-moving Pacific plate, and there is an extensive subduction zone to the north and northwest of it, so there could be more intense mantle flow in that region. So one cannot necessarily generalize from Hawaii.

The numerical model seems to follow established procedures, but I find some of the description a bit problematic. They write their activation energy 146 kJ/mol implies 7 orders of viscosity variations. However, from the first equation of Coltice and Shephard I get $\eta_0(z) \cdot \exp(17079\text{K}/T)$. In order for the exponential to vary by a factor 10^{**7} , the exponent would have to vary by about 16. If one assumes an average potential surface temperature of 1613 K and goes up and down by 300 K (1313 K to 1913 K) the exponent varies between 8.93 and 13.01, so only by about 4 (corresponding to variations by factor ~ 55). To get variations up to 16, one has to go to much lower temperatures, and those only occur in the lithosphere. So those strong viscosity variations can only occur in the lithosphere, which is basically not taking part in the convection; within the convecting mantle they should be less, and that should be made clear. Also, the Rayleigh number (supposed to be 10^{**7}) is inversely proportional to the viscosity value used to calculate it. But if the viscosity varies itself by a factor 10^{**7} , it is not obvious which is the appropriate viscosity to calculate Rayleigh number, so in this case Rayleigh number is not necessarily a meaningful concept.

Since, based on what I wrote until now, I cannot recommend acceptance at this stage, I will not go into further detail in my review. I think the paper could become acceptable, though, if they follow all my recommendations, but this would imply completely changing their conclusions (and title). In this case I would be willing to give a more thorough review.

Bernhard Steinberger

Point-by-point reply to reviewers

Reviewer #2 (Remarks to the Author):

This manuscript provides a novel means to support a general northward drift of the Azores plume since ~85 Ma. The conclusions are based on defining an asymmetric mantle potential temperature profile across the northern Mid-Atlantic ridge then comparing the results to model generated plume-ridge interactions. This provides interesting insights into the Azores plume as well as provides a new method of studying plume-ridge interaction which will be of interest mantle geo-chemists, dynamicists and physicists. This manuscript is suitable for publication in Nature Communication with some revisions.

Major Points

Please segment the manuscript into more discrete sections (e.g. Introduction, Results and Discussion, Conclusion) to improve readability.

We have done that.

The authors argue the 1-2 cm/yr drift of the plume is consistent with the Corner Seamounts being sourced by the Azores plume. However, these seamounts are not dated (to my knowledge) and the ages are modeled based on assuming the seamounts belong to the New England Seamount Chain (Duncan 1984) or backtracked based on sea level erosion (Tucholke and Smoot, 1990). This assumption should be made clear in the manuscript as significant uncertainty remains over the ages of those seamounts. Further discussion of the plate motion – plume motion combination needed to link these Seamounts to the Azores plume is required.

We added the following precision in the manuscript Lines 156-159: “The indirect dating of these complexes¹ was traditionally used to affiliate them to the New-England hotspot track mainly located on the North-American plate². However, recent tectonic³ and geochemical⁴ studies bring complementary information for a genetic link with the Azores.”

Figures need title sentences.

We added titles to figures.

Supplementary figures 3 through 7 and the corresponding impact on the results/discussion use “25% of the dataset”. The authors need to describe what 25% of the data they used and how they filtered/selected the data. Were outlier removed? Was the data filtered at random? Was the entire dataset used in the calculations but only 25% shown in the figures? This has the potential (although I doubt it will change the results) to influence the mantle potential temperature estimates depending on where data was omitted.

Analyses where only 25% of the data are reported (Fig. SI 3, 6 and 7) were conducted on whole rocks. Analyses where 100% of the data are reported (Fig. 2, SI 4, SI 5 and SI 8) include both whole rock and glass analyses. We added figure SI 2a and SI 2b to show that these two datasets are not significantly different on the Mid-Atlantic Ridge. We also changed the sentence “Here, only ~25% of the dataset have been reported” to “Here, only ~25% of the dataset have been reported (whole-rock analyses only, see Fig. SI 2a)” for clarity on corresponding figures in the Supplementary Material.

I feel some mention of the difference in Azores plume motion obtained by your methods (northward) and by mantle flow modeling based on mantle tomography models (Steinberger 2000-JGR) is warranted. Steinberger modeled a Southeast ward drift of the plume since 80 Ma. Although there are uncertainties and assumptions involved in Steinberger (2000)’s method I feel your Northward drift results warrant some comparison.

We agree that we need to discuss differences between our modelling results and the study of Steinberger, 2000, especially since we reach different conclusions. Therefore, we added the following discussion Lines 141-150: “This result differs from previous estimates of the Azores drift from numerical models of instantaneous flow induced by a density distribution based on seismic tomography⁵, which favor a southwestward motion of the plume. However, in such models, the predicted flow close to the core-mantle boundary where the plume moves is very uncertain: (i) in the deepest mantle, tomographic models have relatively low resolution, show significant differences, and the conversion of seismic anomalies to density is ambiguous; (ii) the computed flows do not consider lateral viscosity variations, which are dominant in this area where cold slabs and hot plumes potentially interact; (iii) the ascent of plumes is not computed self-consistently in the flow model contrarily to our models. These models fail to explain both the vertical shape of plumes in the lower mantle recently unraveled by tomographic models⁶, and the southward elongated asymmetry across the Azores since they would predict an asymmetry in the opposite direction.”

On lines 129-132 the authors argue that the observed results from the modelled Ridge axis 2 compares to the modern-day results for the Azores ridge. An addition to figure 4 or new figure 5 comparing the model results of potential mantle temperature for some distance across the ridge directly to the observed data would strengthen this argument. This may not be feasible so I don't require this addition.

We have now changed the results of the modelling section. The observation of the thermal anomaly of the Azores hotspot across the ridge depends on the availability of basalts samples collected away from the MAR. However, such samples are not available to our knowledge.

Line by line points:

Line 36: Azores plume contains low 4He/3He (or high 3He/4He) anomalies as evidence for a more primitive mantle source. The high 4He/3He lava flows (more radiogenic) do not necessarily indicate a deep mantle source.

We have rephrased the sentence to “primitive ⁴He/³He ratios”.

Lines 81 – 83: Seems like a large portion of the data indicate that beneath Azores and Iceland, pyroxene crystallizes in the mantle as well as the crust. Please elaborate or provide appropriate references on how “pyroxenes crystallize only in the oceanic crust”.

We changed the corresponding sentence to: “pyroxenes crystallize mainly in the oceanic crust”.

Lines 88 – 95: Awkward structure/wording.

We changed the corresponding sentence Line 44-51 to: “Theories for interactions between a ridge and a static plume⁷⁻⁹ fail to explain the extent of hot plume material spreading along the ridge (called waist) and its volume flux: accounting for a waist > 2,000 km consistent with bathymetry¹⁰, gravity¹¹, geochemistry¹² (Fig. 2c) and seismic tomography¹³ (Fig. 1) would lead to an unrealistic volume flux of about 400 m³/s¹⁴. The presence of anomalously buoyant plume material more than 2,000 km south of the plume cannot either be explained by a burst of plume activity in the past. Such increase in the volume flux and temperature in the conduit was proposed for the Azores about 10 Ma ago based on V-shaped anomalies¹⁵, but would have led to a local increase of T_p of about 70°C, at about 600 km from the hotspot location and lasting only a few million years⁹.”

Lines 178-202: This read as additional discussion more than methods. I recommend moving it to the start of the supplemental document.

We moved this discussion after Fig. SI 8.

Supp figure 5: I'm confused by the x-axis. I think it should be degree West (?) as Iceland sits at ~19oW. More clarity on what the x-axis represents is required. Do all the MORB samples >~190 East contain MgO contents < 8 wt% or was depth of melting just not calculated for those samples? Please clarify in figure caption.

We changed the figure accordingly. We added the following precision in the figure caption: “The longitude 0° corresponds to the Greenwich meridian”.

Supp figure 6: Where does the crustal age data come from? Reference required.

We added that precision

Pedantic/Minor points:

Inconsistent comma use for integers over 1000 (e.g. line 40 says '3000 km', line 42 uses '3,000 km')

We homogenized the numbers.

Sup Figure 2: Potential is capitalized in the second line.

Done.

Supp figure 5: Third line, there are two parentheses before 'here'.

Done.

Supp figure 5: Below is misspelled as 'Bellow'

Done.

Supp figure 6: Here is misspelled as 'Hhere'

Done.

Figure 3: Crystallization is misspelled on the y-axis of (a).

Done.

Line 157: Awkward wording for the “We tested different...” sentence.

We changed the sentence Line 178 to:” We tested different hypotheses. Results are given in Fig. SI 1.”

Line 115: Space needed between ‘0.38’ and ‘cm/y’
Done.

Thanks for the interesting read,
Kevin Konrad

Reviewer #3 (Remarks to the Author):

This paper supposedly shows evidence for northward drift of the Azores plume in the Earth's mantle. However, I find the argumentation unconvincing and therefore cannot recommend publication. Their main argument is that there is a thermal anomaly associated with the Azores hotspot, and this anomaly stretches further southward than northward. Their conclusion is that the plume must have been further south in the past to cause this thermal anomaly there. This sounds convincing at first sight, and in fact plume-related temperature and lithospheric heat flow anomalies have been proposed elsewhere, but after thinking more carefully, I concluded that it actually cannot be true. This is, because the lithosphere at the ridge didn't exist 85 Myr ago. With a half-spreading rate of the order of 1 cm/yr, the lithosphere that was at the ridge 85 Myr ago is now about 850 km away. Likewise, because of mass conservation, material that is now at the ridge (and causing the higher temperatures) would have been a similar distance away from the ridge back then. So the thermal structure that is there now has likely nothing to do with the location of the plume in the distance past.

In our model, plumes are fully-dynamic, interacting with both the lithosphere and different mantle scales of convection. They are consistent with previous works of Neil Ribe^{7,16}, Maxim Ballmer¹⁷ and other scientists (e.g.^{9,10}) on plume-ridge and plume-plate interactions. Shearing and channelling of plume material generates wakes. It is a problem of advection. So a back of the envelope calculation focused on heat diffusion is not relevant to capture the shape of a temperature anomaly resulting from plume-ridge-plates interaction. The novelty of our models is that they model the full dynamic system of plumes, plates and ridges, in 3D spherical geometry. This has never been done before.

As shown on Figure 3 of the manuscript and Figure rev.1, a plume wake is constantly fed by plume material coming from the conduit and flowing laterally away from the plume over at least 500 km on each of the sides of the ridge axis and at lateral speeds which can reach 10 cm/yr. In the simplest case of a non-moving plume, the material would therefore spread over an ellipsis 100-200 km thick at the base of the lithosphere. Considering that the ridge is continuously fed by material stirred over 200 km from its axis, the thermal anomaly induced by the proximity of the plume material would vanish in about 50 Myr. In the case of a moving plume such as Hotspot 6 moving eastward at a speed of 1 cm/yr, the ellipsis is asymmetric. Its westward end is bevelled as the plume material which was several hundreds of km away from the ridge is now erupting and still has a thermal signature from the former proximity of the plume.

Figure rev.1: (a) Magnitude of the velocity of Hotspot 6 (Case 1) at 100 km depth at 25 Myr. The ridge system is highlighted in red. Orange arrows represent lateral velocities at 100 km depth. White arrows are surface plate velocities. The 1390°C isotherm delimitating the plume wake is contoured in white. **(b)** Interpretative sketch of the lateral motion of a thermal anomaly induced by the Azores plume towards the ridge axis (not to scale). The transparent blue patch is a part of the plume wake (dashed grey contours) initially located 500 km away from the plume centre with a thermal signature of the plume. It is now located at the ridge axis (red lines).

Also, I don't find the numerical experiment convincing to support that case. Supposedly (line 118) the plume moves with 1 cm/yr over 70 My, which would be about 700 km, or 600 km in 60 Myr. However, in Figure 4a I measured that the center of the plume is 2.3 cm from the left margin, and 1.4 cm from the bottom margin. Within my measurement uncertainty, I found it is at the same location also at 40 Myr and at 70 Myr. For the diameter of the Earth I measured 3.8 cm, corresponding to 12742 km. So 1 mm corresponds to 335 km, and a 600 km motion should correspond to nearly 2 mm difference and definitely be visible, but it isn't. Only at 100 Myr, I find the plume slightly ENE at 2.5 cm from the left and 1.45 cm from the bottom margin.

For more clarity and to show more than one situation of a model which displays a diversity of cases, we changed our Figure to one with several cases supporting more clearly our case.

As stated Line 109, the absolute motion of all detected plumes is "1.75±0.38 cm/yr in average".

But the asymmetry of the temperature anomaly is already there at 10, 40 and 70 Myr, so it cannot have been caused by motion of the plume. I find it much more likely that the asymmetry is caused by underlying mantle flow that drags plume material preferentially in one direction. In fact, the plate motions shown would indicate just that to me: the plate to the west shows strong westward motion, and this would presumably correspond to underlying mantle flow in a similar direction.

In our model, we show that the dominant effect acting on the geometry of the plume wake is the shearing caused by the relative direction of motion between the lithosphere and the plume. In the case of the Azores, the plate absolute motions are directed northward (e.g. Torsvik et al., 2010), while the thermal asymmetry is directed southward. Therefore, the only possible explanation is that the plume moves northward relatively to the plates, and faster than them in order to generate an asymmetry (modelled case 2). Our Figure 3 showing all possible cases makes our point clearer.

Similarly, I would argue for the same mechanism working for the Azores hotspot: The closest extensive subduction zones over the past few tens of Myr, corresponding to extensive slabs as seen through tomography, are to the southwest. Hence those sinking slabs would drive southwestward large-scale flow in the direction along the ridge. And in fact geodynamic models based on tomography show just that. So you cannot write in the abstract "we show that the Azores plume moved northward by 1-2 cm/yr from 85 Myr", because this is just your interpretation of results, but I am sure it is wrong.

According to recent seismic tomography models (e.g. Simmons et al., 2010¹⁸), the nearest deep and fast seismic velocity areas are located southwest from the Azores. Considering several tomographic models shows large uncertainties on the location and geometry of those basal seismic anomalies. In this region of the mantle close to the core-mantle boundary, we can also question their interpretation in terms of density anomalies (i.e. what is the effect of composition vs. temperature). So first, the density model in this area at the base of the mantle is very uncertain.

Instantaneous mantle flow models based on the conversion of seismic anomalies into density anomalies generate a southward motion^{5,19}, and produce an important tilt of the Azores plume⁵. According to tomographic models unravelling the structure of mantle plumes⁶, the structure of mantle plumes is quasi vertical, at least in the lower mantle. So there is a contradiction here. It is not surprising because the models used to predict plume motions do not model plumes in a fully dynamic way, and do not take into account viscosity variations caused by temperature. Close to the CMB, temperature contrasts are strong and generate viscosity variations of several orders of magnitude. Our model captures that and produces nearly vertical plumes that rise at larger velocities than the ones assumed in instantaneous models to advect plume material upward (Figure SI 9).

Therefore, we think our approach is more robust for the Azores plume, since we use a combination of data which are better constrained: upper mantle seismic models, geochemistry, petrology and bathymetry. Also we are modelling the plume flow in a self-consistent manner and obtain Earth's like characteristics for them.

Also, I find the interpretation of observational data unconvincing. Not that I would understand any of this (as I'm not a geochemist) but for example you conclude from Figure 3b a decrease by 0.03°C/km towards the south and 0.12°C/km (on line 70) or even 0.8°C/km (on line 131) towards the north. But as far as I can see, this is just eyeballing, and no proper statistics is being done.

We have changed the way we calculate the asymmetry of the thermal anomaly for simplicity and more clarity. Hence, we now calculate a symmetry coefficient corresponding to the ratio between the northern extent of the thermal anomaly and the southern extent, on each side of the Azores (Fig. 3f). The major point here is that the thermal anomaly related to the Azores extends in one direction (South). This is the clearest observation. The precise values of the thermal gradients are not fundamental.

Also, they interpret the Corner rise and Great Meteor oceanic plateaus as part of the Azores hotspot track.

However, the usual interpretation (e.g. Tucholke and Smoot) is that the Great Meteor Seamount is at the young end of a hotspot track that also includes seamounts to the north of it. This is also supported by relatively young radiometric ages from Great Meteor seamount, indicating that it was formed ~20 Myr ago. Also, this trend fits e.g. the plate motion shown in Figure 8 SI. So I find the "traditional" interpretation -- that Great Meteor is a separate hotspot (perhaps now extinct) that first created a track on the North American plate (starting in New England, followed by New England and Corner Seamounts, then was overridden by the ridge, and subsequently created a track on the African plate, with its young end to the South) much more convincing and compatible with observations.

We discussed this point by adding Line 156-159: "The indirect dating of these complexes¹ was traditionally used to affiliate them to the New-England hotspot track mainly located on the North-American plate². However, recent tectonic³ and geochemical⁴ studies bring complementary information for a genetic link with the Azores."

Also, they argue that there is evidence that the Hawaii hotspot has moved, and likewise, the Azores hotspot should also have moved. However, one can find perfectly good reasons why the Hawaii hotspot should have moved much more: Because it is beneath the fast-moving Pacific plate, and there is an extensive subduction zone to the north and northwest of it, so there could be more intense mantle flow in that region. So one cannot necessarily generalize from Hawaii.

We agree that we cannot generalize from possible mechanisms of the motion of the Hawaiian plume. We just note that the only plume with a clear evidence of motion is Hawaii. For this plume, several causes have been invoked to explain its southward drift (ridge capture²⁰ or slab push and LLSVP deformation¹⁹). Our study does not provide an explanation for the northward drift of the Azores but states that such motion is necessary to explain the asymmetry observed along the ridge from petrology, seismology, geochemistry and bathymetry. Such drift is consistent with recent geochemical⁴ and tectonic data³ making direct links with Great Meteor. We open the discussion Lines 166-171 by proposing that: "The possible anchorage of the Azores plume on the edge of a dense and stable large low shear velocity province (LLSVP), as suggested in tomographic models²¹, is not accounted by our convection model. However, convection calculations do not show enhanced fixity of plumes with chemical anomalies at the core-mantle boundary²². Also, slow deformation of the edges of LLSVPs produce plume motions of the order of 1 cm/yr²³, and plumes can propagate longitudinally along the edges of LLSVPs which undergo significant thermomechanical or thermochemical instabilities²⁴."

The numerical model seems to follow established procedures, but I find some of the description a bit problematic. They write their activation energy 146 kJ/mol implies 7 orders of viscosity variations. However, from the first equation of Coltice and Shephard I get $\eta_0(z) \cdot \exp(17079K/T)$.

In order for the exponential to vary by a factor 10^{**7} , the exponent would have to vary by about 16. If one assumes an average potential surface temperature of 1613 K and goes up and down by 300 K (1313 K to 1913 K) the exponent varies between 8.93 and 13.01, so only by about 4 (corresponding to variations by factor ~55). To get variations up to 16, one has to go to much lower temperatures, and those only occur in the lithosphere. So those strong viscosity variations can only occur in the lithosphere, which is basically not taking part in the convection; within the convecting mantle they should be less, and that should be made clear. Also, the Rayleigh number (supposed to be 10^{**7}) is inversely proportional to the viscosity value used to calculate it. But if the viscosity varies itself by a factor 10^{**7} , it is not obvious which is the appropriate viscosity to calculate Rayleigh number, so in this case Rayleigh number is not necessarily a meaningful concept.

The 7 orders of viscosity variations include the lithosphere. We have clarified this point by precisizing Line 208-209: "The activation energy is 146 kJ/mol, implying 7 orders of viscosity variations throughout the domain.". Our Rayleigh number is appropriate with the description we make of the model. This is the Rayleigh number definition in our system of equations. Anyway, the Rayleigh number of the Earth is not known, and what is the most relevant here is that our model produces values of heat flow, plate velocities, plume temperature excess and buoyancy fluxes falling in the estimates of the Earth (Fig. SI 9). This has never been done before with 3D spherical convection with plate-like behaviour.

Since, based on what I wrote until now, I cannot recommend acceptance at this stage, I will not go into further detail in my review. I think the paper could become acceptable, though, if they follow all my recommendations, but this would imply completely changing their conclusions (and title). In this case I would be willing to give a more thorough review.

Bernhard Steinberger

References:

1. Tucholke, B. E. & Smoot, N. C. Evidence for Age and Evolution of Comer Seamounts and Great Meteor Seamount Chain From Multibeam Bathymetry. *J. Geophys. Res.* **95**, 17555–17569 (1990).
2. Müller, R. D., Royer, J.-Y. & Lawver, L. A. Revised plate motions relative to the hotspots from combined Atlantic and Indian Ocean hotspot tracks. *Geology* **21**, 275–278 (1993).
3. Gente, P., Maia, M. & Goslin, J. Interaction between the Mid-Atlantic Ridge and the Azores hot spot during the last 85 Myr: Emplacement and rifting of the hot spot-derived plateaus. *Geochemistry Geophys. Geosystems* **4**, 1–23 (2003).
4. Pinto Ribeiro, L., Martins, S., Hildenbrand, A., Madureira, P. & Mata, J. The genetic link between the Azores Archipelago and the Southern Azores Seamount Chain (SASC): The elemental , isotopic and chronological evidences. *Lithos* **295**, 133–146 (2017).
5. Steinberger, B. Plumes in a convecting mantle: Models and observations for individual hotspots. *J. Geophys. Res.* **105**, 11127–11152 (2000).
6. French, S. W. & Romanowicz, B. Broad plumes rooted at the base of the Earth’s mantle beneath major hotspots. *Nature* **525**, 95–99 (2015).
7. Ribe, N. M., Christensen, U. R. & Theiging, J. The dynamics of plume-ridge interaction , 1 : Ridge-centered plumes. **134**, 155–168 (1995).
8. Ito, G., Shen, Y., Hirth, G. & Wolfe, C. J. Mantle flow, melting, and dehydration of the Iceland mantle plume. *Earth Planet. Sci. Lett.* **165**, 81–96 (1999).
9. Albers, M. & Christensen, U. R. Channeling of plume flow beneath mid-ocean ridges. *Earth Planet. Sci. Lett.* **187**, 207–220 (2001).
10. Ito, G. & Lin, J. Oceanic spreading center– hotspot interactions: Constraints from along-isochron bathymetric and gravity anomalies. *Geology* 657–660 (1995).
11. Thibaud, R., Gente, P. & Maia, M. A systematic analysis of the Mid-Atlantic Ridge morphology and gravity between 15°N and 40°N: constraints on the thermal structure. *J. Geophys. Res.* **103**, 24,223–24,243 (1998).
12. Dosso, L., Bougault, H., Langmuir, C., Bollinger, C. & Bonnier, O. The age and distribution of mantle heterogeneity along the Mid-Atlantic Ridge (31 – 41°N). *Earth Planet. Sci. Lett.* **170**, 269–286 (1999).
13. Debayle, E., Dubuffet, F. & Durand, S. An automatically updated S -wave model of the upper mantle and the depth extent of azimuthal anisotropy. *Geophys. Res. Lett.* (2016). doi:10.1002/2015GL067329.1.
14. Ito, G., Lin, J. & Graham, D. Observational and theoretical studies of the dynamics of mantle plume-mid ocean ridge interaction. *Rev. Geophys.* **41**, 1017 (2003).
15. Cannat, M. *et al.* Mid-Atlantic Ridge – Azores hotspot interactions : along-axis migration of a hotspot-derived event of enhanced magmatism 10 to 4 Ma ago. *Earth Planet. Sci. Lett.* **173**, 257–269 (1999).
16. Ribe, N. M. & Delattre, W. L. The dynamics of plume-ridge interaction - III . The effects of ridge migration. *Geophys. J. Int.* **133**, 511–518 (1998).
17. Ballmer, M. D., Ito, G., Hunen, J. Van & Tackley, P. J. Spatial and temporal variability in Hawaiian hotspot volcanism induced by small-scale convection. *Nat. Geosci.* **4**, 457–460 (2011).
18. Simmons, N. A., Forte, A. M., Boschi, L. & Grand, S. P. GyPSuM: A joint tomographic model of mantle density and seismic wave speeds. *J. Geophys. Res.* **115**, (2010).
19. Hassan, R., Müller, R. D., Gurnis, M., Williams, S. E. & Flament, N. A rapid burst in hotspot motion through the interaction of tectonics and deep mantle flow. *Nature* **533**, 239–242 (2016).
20. Tarduno, J., Bunge, H., Sleep, N. & Hansen, U. The Bent Hawaiian-Emperor Hotspot Track : Inheriting the Mantle Wind. *Science (80-.)*. **324**, 50–54 (2009).
21. Thorne, M. S. & Garnero, E. J. Inferences on ultralow-velocity zone structure from a global analysis of SPdKS waves. *J. Geophys. Res.* **109**, 1–22 (2004).
22. McNamara, A. K. & Zhong, S. The influence of thermochemical convection on the fixity of mantle plumes. *Earth Planet. Sci. Lett.* **222**, 485–500 (2004).
23. Flament, N., Williams, S., Müller, R. D., Gurnis, M. & Bower, D. J. Origin and evolution of the deep thermochemical structure beneath Eurasia. *Nat. Commun.* (2017). doi:10.1038/ncomms14164
24. Ni, S., Tan, E., Gurnis, M. & Helmberger, D. Sharp Sides to the African Superplume. *Science (80-.)*. **296**, (2002).

Reviewers' comments:

Reviewer #2 (Remarks to the Author):

Review of the revised Arnould et al. (2019) - Northward drift of the Azores plume in the Earth's mantle

This revised manuscript is significantly improved from the early version. I am glad to see it resubmitted to Nature Communications as I feel it is appropriate for this journal. I recommend publishing the article after minor revisions.

Line by line points:

Main Text

Line 29: Area should be 150,000 km²

Line 38-39: North and south should not be capitalized here.

Line 39 and Figure 2d: Neither Nb nor Zr are rare earth elements. So the use of 'REE ratios' for Nb/Zr is wrong. Please change to something like 'key trace element ratios'. Also, I recommend mentioning in the figure caption they are normalized to the chondrite values in Bougault and Treuil (1980). Those chondrite values are out-of-date but it's of no consequence to your figure.

Line 55: 'plumes' should not be plural here.

Supplements

The author affiliations are different between the supplements and the main text.

Supplement overview is incomplete on the title page. Missing "II – Analysis of the thermal imprint of modeled plumes"

Table SI 1: A brief overview (2-3 sentences) of what the table covers and what the columns represent would be appreciated.

Thanks and good luck,
Kevin Konrad

Reviewer #3 (Remarks to the Author):

I still don't find the evidence for northward drift of the of the Azores plume convincing, and below I will lay out in detail why. However, it is one possible interpretation, and at this point I support publication -- as long as appropriate caveats are given -- although I disagree with the interpretation of the authors. In this way, the paper may contribute to the discussion about hotspot motion.

Essentially, in order to get the kind of wake which the authors argue for, one needs a motion of hotspot relative to the plates northward or northeastward along the ridge, but this could be due to either north- to northeastward hotspot motion or a south-to southwestward component of absolute plate motions. The plate motions which the authors show in Figure 1 have a northward component, but one has to bear in mind that these are slow and directions have large uncertainties and differ between different reference frames. The authors write that Steinberger et al., O'Neill et al. and Torsvik et al. (2008,2010) reconstruct a northward motion at a speed >1cm/yr, but this is actually only true for the African hotspot reference frame of O'Neill et al. and adopted by Torsvik et al. (2010) which is shown in Figure 1a. The Global Moving Hotspot Reference frame of Steinberger et al. (2004) and adopted by Torsvik et al. (2008) has an Africa rotation pole at 29 N, 27.5 W, close to Azores i.e. predicts a very slow (and westward) Africa plate motion of only a few mm/yr at Azores. This reference

frame was updated by Doubrovine et al. (2012) who obtained a rotation pole at 37 N, 33 W, again leading to very slow plate motion at Azores. I find a global moving hotspot reference frame more reliable (at least for the past ~47 Myrs, when there are no serious plate circuit issues), because it is based on a larger number of hotspots. However, more importantly, with the African plate essentially stationary in that region, the Mid-Atlantic Ridge would have overridden the Azores hotspot ~40 Myr ago, so the earlier in time, the more relevant the North American plate motion would be, which, in that reference frame, moves ~2 cm/yr westward. So it has a component along the part of the ridge which runs southwestward from the Azores, but not along the part of the ridge which runs northward. Hence, the asymmetric wake could be explained due to plate motion, not plume motion.

Furthermore, I am not sure whether the plume volume flux of about $400 \text{ m}^3/\text{s}$ which the authors write would be necessary to explain a waist $>2000 \text{ km}$ is necessarily unrealistic. It is true that current estimates are about $51\text{-}90 \text{ m}^3/\text{s}$, so this would be about a factor 5-8 higher. But then previous estimates for the Iceland plume were about the same, whereas Parnell-Turner (Nature Geoscience volume 7, pages 914–919, 2014) estimate a flux more than 10 times higher. I am not aware of similar estimates for Azores, but this makes me wonder whether there aren't also substantial uncertainties.

Also, I find much of the observational evidence given not very convincing. I won't comment on the new evidence reported here, because this is not my field of expertise, but discuss the previous evidence given: From Figure 2c, I find the topography decline 500 km northward very similar to 500 km south(west) ward. A bit further away, topography actually goes up again towards the north, which could be interpreted as stronger influence towards the north. Then the profile is shown further towards the south, so that part cannot be compared with the north, which, at latitudes further north, is presumably influence by the Iceland plume. Likewise, the Bouguer gravity profile looks more or less symmetric to me for those parts that can be compared. They show one particular tomography model which indicates stronger influence towards the south, but I find it can be quite misleading to just look at one model, if there are many of them, with quite a variety between them (see e.g. the "SubMachine" website). For example, I find any asymmetry far less obvious with the SL2013sv model, and the LLNL-63Dv3 model would actually indicate more influence towards the north. Also, the regional tomography model mentioned on line 42 doesn't show it, because it doesn't even have the extent and data coverage of 3000 km.

Also, they cite V-shaped structures detected by Cannat et al. as evidence, but those authors actually give a different explanation -- along-axis migration of a hotspot-derived event of enhanced magmatism, similar to the explanation originally given by Vogt for Iceland. Also, I still find it rather problematic to link Great Meteor Seamount to Azores. The radiometric ages for this seamount have been determined at 11-16 Ma, and I would consider this rather "hard" evidence in contrast to the tectonic and geochemical studies enlisted here.

In the following, a few more consecutive comments:

Abstract lines 23-24 "we show that ... is a consequence of the Azores plume moving northward by 1-2 cm/yr" -- As explained above, I find this a rather bold statement, and I would ask the authors to tone it down to accommodate dissenting views like mine. Similarly on line 58 "only a northward motion of the plume $> 1 \text{ cm/yr}$ can explain" - I don't agree that there are no alternative explanations.

Line 128: "Fig. 3f ... show that temporal changes" -- sorry but I don't see anything related to temporal changes in Fig. 3f.

Then you criticize my work, but I can equally name a few things where my approach is better than yours. One thing is that I can match the actual position of plumes, in fact I think my approach is the only way developed so far how you can model hotspot motion in a realistic large-scale flow and match the actual hotspot positions. Also, my large-scale flow models have various constraints, e.g. from geoid, and global total heat flux and I don't see any constraints for yours. Did you compute the core-mantle-boundary and surface heat flux from your models; is it realistic? The fact that the total heat

flux inferred from plumes buoyancy fluxes is much lower than estimates for what the geodynamo requires has also been used as an argument that plume flux is perhaps severely underestimated (see my comment above). And your plume fluxes in the lower mantle are even much lower than near the surface. Of course, some of the heat from the core might also be carried away in large-scale flow, not just through plumes.

Then you criticize my work, but I can equally name a few things where my approach is better than yours. One thing is that I can match the actual position of plumes, in fact I think my approach is the only way developed so far how you can model hotspot motion in a realistic large-scale flow and match the actual hotspot positions. Also, my large-scale flow models have various constraints, e.g. from geoid, and global total heat flux and I don't see any constraints for yours. Did you compute the core-mantle-boundary and surface heat flux from your models; is it realistic? The fact that the total heat flux inferred from plumes buoyancy fluxes is much lower than estimates for what the geodynamo requires has also been used as an argument that plume flux is perhaps severely underestimated (see my comment above). And your plume fluxes in the lower mantle are even much lower than near the surface. Of course, some of the heat from the core might also be carried away in large-scale flow, not just through plumes.

Line 151: "unravalled by tomography models" -- it is actually not (plural) tomography model"s" but only the one by French and Romanowicz which shows vertical plumes in the lower mantle. Besides, this model doesn't even clearly show the Azores plume (only "somewhat resolved") and if, with some good will you see an Azores plume in a cross section (e.g. again with the Submachine website) it is actually also tilted, coming up from the south, similar to my models. Also, Nelson and Grand (Nature Geoscience, 2018) show a Yellowstone plume conduit that is tilted similarly to my model predictions. So it is a misrepresentation if you write that my models "fail to explain the vertical shape of plumes in the lower mantle recently unraveled by tomographic models".

References have mistakes. I didn't check thoroughly, but here are a few I found

#2: Third author should be "Redfield"

#4 and 31: Why "Science (80-.)"?

#20 authors Solomon and Hung are missing

(besides, I would rather write "Van der Voo" and "Van der Lee" as last names in #4 and #20)

#27 write "McKenzie" with capital K.

#40 Miller needs to be fixed.

Supplementary #6: "convection models" is missing.

Besides, make it consistent which letters are capitalized.

Figures in general: it would be useful and make the reading easier if there was also a scale, or at least tickmarks, on the right hand side.

I find a reference viscosity of 1.03×10^{22} Pas, presumably used for the upper mantle, much too high. It is about 10 times higher than the classical "Haskell average" inferred from postglacial rebound, and much higher of other estimates of viscosity below the lithosphere. Also, if the lower mantle viscosity is 30 times higher, it would be about 3×10^{23} Pas, which I also consider much too high. I may interpret your numbers wrong, but if so, that would need clarification. With such (at least in my understanding) unrealistic values, I find it actually quite surprising that your results, at least in terms of plume size and spacing, look quite realistic, similar to, for example Hassan et al. (G-cubed, 2015) who use, in my view, a much more realistic viscosity structure. What I find a bit hard to understand, though, is why your plumes have a volume flux and heat flux about a factor ~ 8 lower in the lower mantle than in the upper mantle. As I understand your equations, your computations are incompressible, so the plume volume flux should remain the same through the mantle, simply because of mass conservation (what flows in in the bottom should come out in the top). Of course, there may be some entrainment, but increase by a factor ~ 8 appears quite unrealistic to me.

One-by-one point response to reviewers

Reviewer #2 (Remarks to the Author): Review of the revised Arnould et al. (2019) - Northward drift of the Azores plume in the Earth's mantle

This revised manuscript is significantly improved from the early version. I am glad to see it resubmitted to Nature Communications as I feel it is appropriate for this journal. I recommend publishing the article after minor revisions.

Line by line points:

Main Text

Line 29: Area should be 150,000 km²

Changed accordingly.

Line 38-39: North and south should not be capitalized here.

OK

Line 39 and Figure 2d: Neither Nb nor Zr are rare earth elements. So the use of 'REE ratios' for Nb/Zr is wrong. Please change to something like 'key trace element ratios'. Also, I recommend mentioning in the figure caption they are normalized to the chondrite values in Bougault and Treuil (1980). Those chondrite values are out-of-date but it's of no consequence to your figure.

Changed accordingly.

Line 55: 'plumes' should not be plural here.

OK

Supplements

The author affiliations are different between the supplements and the main text.

Changed accordingly.

Supplement overview is incomplete on the title page. Missing "II – Analysis of the thermal imprint of modeled plumes"

Changed accordingly.

Table SI 1: A brief overview (2-3 sentences) of what the table covers and what the columns represent would be appreciated.

We added information about the content of each column in the corresponding table and we added the corresponding sentences in the supplementary caption of Table SI 1: "Solutions of our calculation are provided in supporting information Table 1a and 1b S1. Table 1a shows the MORBs compositions along the MAR, their mantellic origin (peridotite vs pyroxenite) and the pressures of crystallization of magmatic pyroxenes (fractionation). Table 1b shows the calculation results from PRIMELT."

Thanks and good luck,
Kevin Konrad

Reviewer #3 (Remarks to the Author):

I still don't find the evidence for northward drift of the of the Azores plume convincing, and below I will lay out in detail why. However, it is one possible interpretation, and at this point I support publication -- as long as appropriate caveats are given -- although I disagree with the interpretation of the authors. In this way, the paper may contribute to the discussion about hotspot motion.

Essentially, in order to get the kind of wake which the authors argue for, one needs a motion of hotspot relative to the plates northward or northeastward along the ridge, but this could be due to either north- to northeastward hotspot motion or a south-to southwestward component of absolute plate motions. The plate motions which the authors

show in Figure 1 have a northward component, but one has to bear in mind that these are slow and directions have large uncertainties and differ between different reference frames. The authors write that Steinberger et al., O'Neill et al. and Torsvik et al. (2008,2010) reconstruct a northward motion at a speed $>1\text{cm/yr}$, but this is actually only true for the African hotspot reference frame of O'Neill et al. and adopted by Torsvik et al. (2010) which is shown in Figure 1a. The Global Moving Hotspot Reference frame of Steinberger et al. (2004) and adopted by Torsvik et al. (2008) has an Africa rotation pole at 29 N, 27.5 W, close to Azores i.e. predicts a very slow (and westward) Africa plate motion of only a few mm/yr at Azores. This reference frame was updated by Doubrovine et al. (2012) who obtained a rotation pole at 37 N, 33 W, again leading to very slow plate motion at Azores. I find a global moving hotspot reference frame more reliable (at least for the past ~ 47 Myrs, when there are no serious plate circuit issues), because it is based on a larger number of hotspots. However, more importantly, with the African plate essentially stationary in that region, the Mid-Atlantic Ridge would have overridden the Azores hotspot ~ 40 Myr ago, so the earlier in time, the more relevant the North American plate motion would be, which, in that reference frame, moves $\sim 2\text{ cm/yr}$ westward. So it has a component along the part of the ridge which runs southwestward from the Azores, but not along the part of the ridge which runs northward. Hence, the asymmetric wake could be explained due to plate motion, not plume motion.

We chose Torsvik et al., 2010 Africa's moving hotspot reference frame, based on the O'Neill et al., 2005's one, since hotspot motions derive from a geodynamic model which accounts for uncertainties in both hotspot reconstructions and mantle convection models.

The models of Steinberger et al., 2004 and Torsvik et al., 2008 predict a westward motion of Africa of 2.7 mm/yr in the vicinity of the Azores. Hence, in both models, only a northward motion of the Azores plume is able to explain the southward-oriented thermal anomaly beneath the Azores, since the African plate is rather stationary.

Doubrovine et al., 2012's model shows little motion of the African and European plates and a westward motion of the North-American plate of 2 cm/yr (the North-South component of its motion is less than 4 mm/yr). The goal of this kinematic reconstruction was to obtain a perfect fit to hotspot tracks worldwide. The major adjusting parameters are plume motion and net rotation. As a consequence, both are the largest among plate reconstruction models (Williams et al., 2015, Torsvik and Steinberger, 2010). Therefore, we consider this reference frame as an end-member. In this reference frame, considering that the Azores hotspot track follows the ridge axis and consists of the Corner Rise and Great Meteor seamounts (as presented in our manuscript), plate-plume kinematics would correspond to our Case 3 (Fig. 3c): accounting for a stationary plume or an eastward moving plume would lead to a westward-oriented wake, generating a symmetric thermal anomaly along the ridge (as shown on Fig. 3e), which does not correspond to the geometry of the observed thermal anomalies beneath the MAR. Therefore, only a northward drift of the Azores plume would generate an asymmetry. If we consider the Azores hotspot track is north-west of the Azores as considered in the Doubrovine et al., 2012's study, then it fails to explain the associated observed thermal asymmetry beneath the MAR. Another mechanism to generate it should be investigated.

To account for this remark, we changed the manuscript by adding Line 136-137: "Therefore, we compared the observed thermal asymmetry with the motion of the African, European and North-American plates in several reference frames based on moving hotspots." and Line 139-147: "Using a global moving hotspot reference frame, Steinberger et al.³⁸ and Torsvik et al.³⁹ proposed a stable African plate. In that case, only a northward motion of the Azores plume can explain southward-oriented thermal asymmetry. Doubrovine et al.⁴⁰ reconstructed both a stationary Africa and Europe and a westward-drifting North-America by 2 cm/yr (case 3, Fig. 3c). Accounting for a stationary or eastward-moving plume would generate a westward-oriented asymmetric wake which would result in a symmetric thermal anomaly along the MAR axis (Fig. 3e). Therefore, only a northward motion of the Azores plume is consistent with the observed thermal asymmetry. In the moving hotspot reference accounting for Indo-Atlantic hotspot tracks, O'Neill⁴¹ and Torsvik et al.⁴² reconstructed a northward motion of Africa at a speed larger than 1 cm/yr in the vicinity of the Azores."

Furthermore, I am not sure whether the plume volume flux of about $400\text{ m}^3/\text{s}$ which the authors write would be necessary to explain a waist $>2000\text{ km}$ is necessarily unrealistic. It is true that current estimates are about $51\text{-}90\text{ m}^3/\text{s}$, so this would be about a factor 5-8 higher. But then previous estimates for the Iceland plume were about the same, whereas Parnell-Turner (Nature Geoscience volume 7, pages 914–919, 2014) estimate a flux more than 10 times higher. I am not aware of similar estimates for Azores, but this makes me wonder whether there aren't also substantial uncertainties.

Indeed, estimates of the volume flux of mantle plumes show uncertainties and different methods (such as the ones of Sleep, 1990, Crosby and McKenzie 2009 or Parnell-Turner et al., 2014 (using V-shaped ridges)) show different values, for example for the Iceland plume. Although Cannat et al., 1999 showed the existence of V-shaped ridges for the Azores plume, they did not estimate corresponding volume flux variations, which could probably be higher than existing estimates from other methods. Therefore, we changed the sentence Line 48 to: "accounting for a waist $> 2,000\text{ km}$ consistent with bathymetry⁷, gravity¹¹, geochemistry¹² (Fig.

2c) and seismic tomography¹⁹ (Fig. 1) would lead to a volume flux of about 400 m³/s²⁶, at least 5 times greater than current estimates for the Azores plume^{20,21}.”

Also, I find much of the observational evidence given not very convincing. I won't comment on the new evidence reported here, because this is not my field of expertise, but discuss the previous evidence given: From Figure 2c, I find the topography decline 500 km northward very similar to 500 km south(west) ward. A bit further away, topography actually goes up again towards the north, which could be interpreted as stronger influence towards the north. Then the profile is shown further towards the south, so that part cannot be compared with the north, which, at latitudes further north, is presumably influenced by the Iceland plume. Likewise, the Bouguer gravity profile looks more or less symmetric to me for those parts that can be compared.

In the vicinity of the Azores, several studies describe the asymmetry of topography and gravity as similar to the geochemical variations (e.g. Dosso et al., 1999, Silveira et al., 2006). The association of a variety of consistent signals lets little room for discarding asymmetry. If we would have considered one of the presented observations isolated from others as an evidence for the asymmetry, it would have been incomplete. For example, taken in isolation, the geoid anomaly of the Azores lies on the flank of a larger-scale geoid high (Detrich et al., 1995, Bowin et al., 1984), which makes it more difficult to interpret alone. Gravity anomalies are also affected by the crustal thickness, which varies along the ridge (e.g. Gente et al., 2003). Hence, we think that the combination of topography, gravity anomalies, geochemistry, seismic tomography and potential temperature data, taken together and showing similar distributions along the ridge, are a pertinent evidence for thermal asymmetry beneath the MAR in the vicinity of the Azores.

They show one particular tomography model which indicates stronger influence towards the south, but I find it can be quite misleading to just look at one model, if there are many of them, with quite a variety between them (see e.g. the "SubMachine" website). For example, I find any asymmetry far less obvious with the SL2013sv model, and the LLNL-63Dv3 model would actually indicate more influence towards the north. Also, the regional tomography model mentioned on line 42 doesn't show it, because it doesn't even have the extent and data coverage of 3000 km.

We have chosen the model 3D2017_09SV of Debayle et al., 2016 since it is the newest available version of a high-resolution surface wave model of the upper mantle, updating automatically. Therefore, it has enough resolution to resolve uppermost mantle structures. We compared this model with other S-wave and surface wave models at 100 km depth (Fig.1 here-below). P-wave tomography has a lower resolution in the shallow mantle and should be interpreted with care. The vote map for 19 S-wave models shows that the asymmetric pattern of low seismic anomalies extending southward the Azores is consistent and extends over 3000 km.

Fig. 1: Low-seismic (-2/+2 dVS) velocity vote map at a depth of 100 km of 17 S-wave models and the depth-restricted 3D2016_09Sv and SL2013sv models available on the Submachine website (Hosseini et al, 2018, Shephard et al, 2017).

Meanwhile, we are also very careful not to draw conclusions directly from tomography in terms of potential temperature of the mantle below the MAR (since slow seismic anomalies in the uppermost mantle could be interpreted by both high temperatures and/or chemical heterogeneities), which is why we investigate the potential temperature of the mantle via the PRIMELT method. The resulting T_P happens to show a very good

agreement with the vast range of S-wave and surface-wave seismic tomography models shown on Fig. 1 here-above.

We have also changed the reference 42 to the one of Pilidou et al., 2005, which is a Rayleigh wave tomography model of the upper mantle for the North-Atlantic area. This model presents an elongated slow seismic anomaly extending southwards between 75 and 150 km at least. Likewise, the authors interpreted the asymmetry of the seismic anomaly as a manifestation of hotspot-ridge interaction.

Also, they cite V-shaped structures detected by Cannat et al. as evidence, but those authors actually give a different explanation -- along-axis migration of a hotspot-derived event of enhanced magmatism, similar to the explanation originally given by Vogt for Iceland.

We cited the V-shaped ridges of Cannat et al., 1999 to show that the large extent of the thermal anomaly south the Azores cannot be accounted by a pulse of magmatic activity of the scale of the ones causing V-shaped ridges, since their age, speed and thermal decay are inconsistent with a present-day extent of more than 2,000 km south the Azores. To make our point clearer, we modified the text on Line 48-52: "The presence of anomalously buoyant plume material more than 2,000 km south of the plume cannot either be explained by the burst of plume activity 10 Ma ago¹⁸, suggested by the existence of V-shaped anomalies south of the Azores. This would only have led to a local increase of TP of about 70°C, at about 600 km from the hotspot location and lasting only a few million years²⁵."

Also, I still find it rather problematic to link Great Meteor Seamount to Azores. The radiometric ages for this seamount have been determined at 11-16 Ma, and I would consider this rather "hard" evidence in contrast to the tectonic and geochemical studies enlisted here.

We agree that existing K-Ar ages of Great Meteor correspond to 11-16 Ma (based on 2 basalt samples, Wendt et al., 1976). However, as written by the authors, both samples are altered. "Consequently, the apparent ages may be diminished by argon losses and partly also by gain of potassium. Therefore, the basalt ages [...] should be regarded as *minimum* ages." Moreover, the samples are characterized by a lack of glass and a high vesicularity, and they correspond to the last volcanic events in the Great Meteor seamounts (Gente et al., 2003).

Recently, Ribeiro et al. 2017, provided K-Ar ages of 31-33 Myr for two seamounts of Great Meteor.

In contrast, indirect dating by Verhoef, 1984 based on the elastic thickness of the lithosphere reveal ages ranging between 65 and 38 Ma and the one of Tucholke and Smoot, 1990, based on the subsidence of Corner Rise and Great Meteor seamounts suggest ages comprised between 75 and 21 Ma for the Great Meteor complex and 86 and 50 Ma for Corner Rise.

Considering such age uncertainties of the Great Meteor seamounts, Gente et al., 2003 proposed an alternative way of dating these seamounts, based on tectonic arguments.

In order to account for such different ages and ways of dating great Meteor, we added Line 166-169: "The direct K-Ar⁴⁶ and indirect⁴⁷ dating of these complexes were traditionally used to affiliate them to the New-England hotspot track mainly located on the North-American plate⁴⁸, despite major uncertainties on the determination of age of these seamounts (ranging from 11 to 65 Ma for Great Meteor)."

In the following, a few more consecutive comments:

Abstract lines 23-24 "we show that ... is a consequence of the Azores plume moving northward by 1-2 cm/yr" -- As explained above, I find this a rather bold statement, and I would ask the authors to tone it down to accommodate dissenting views like mine. Similarly on line 58 "only a northward motion of the plume > 1 cm/yr can explain" - I don't agree that there are no alternative explanations.

As stated in the first comment here-above, we show that, irrespective of the reference frame, the geometry of the observed thermal asymmetry is only consistent with a northward drift of the Azores plume, provided that its hotspot track is located southward the Azores plateau and comprises the Great Meteor and Corner Rise seamounts. A hotspot track located north-west of the plateau would be inconsistent with the observed geometry of the thermal anomaly south of the Azores and would require a different mechanism. Therefore, we have chosen not to change our statements.

Line 128: "Fig. 3f ... show that temporal changes" -- sorry but I don't see anything related to temporal changes in Fig. 3f.

We have added an arrow on Fig. 3f showing that the points corresponding to Case 3 correspond to a change from being less symmetric to symmetric then to asymmetric again in 55 Ma. The full figure corresponding to this temporal evolution of the asymmetry of the corresponding plume is shown of Fig. SI 11.

Then you criticize my work, but I can equally name a few things where my approach is better than yours. One thing is that I can match the actual position of plumes, in fact I think my approach is the only way developed so far how you can model hotspot motion in a realistic large-scale flow and match the actual hotspot positions. Also, my large-scale flow models have various constraints, e.g. from geoid, and global total heat flux and I don't see any constraints for yours. Did you compute the core-mantle-boundary and surface heat flux from your models; is it realistic? The fact that the total heat flux inferred from plumes buoyancy fluxes is much lower than estimates for what the geodynamo requires has also been used as an argument that plume flux is perhaps severely underestimated (see my comment above). And your plume fluxes in the lower mantle are even much lower than near the surface. Of course, some of the heat from the core might also be carried away in large-scale flow, not just through plumes.

We agree that our models cannot match the actual position of plumes with our method, by design: our models produce self-consistently evolving convective flows and plate tectonics, and are not built to fit kinematic observations. The purpose of these sentences is to highlight the differences between the two approaches, which can be seen as complementary. We added the following point Line 153-154: “Although such models have matched the actual position of plumes on Earth, the predicted flow close to the core-mantle boundary where the plume moves is uncertain”.

We have information about the CMB and surface heat flux of our model. We added it in the Supplementary section, in the numerical set-up section: “This model presents an average basal heat flow of 6 TW, which corresponds to 17% of the total surface heat flow (34 TW)”. The total heat transported by plumes to the surface corresponds to about 30% of the total heat flow from the core. This small contribution corresponds to heat flux estimates from numerical models of incompressible and isoviscous convection (Labrosse, 2002), although compressibility, plate tectonics, large-scale mantle convection or internal heating modes may alter those conclusions (e.g. Gonnermann et al., 2004).

Line 151: "unravalled by tomography models" -- it is actually not (plural) tomography model"s" but only the one by French and Romanowicz which shows vertical plumes in the lower mantle. Besides, this model doesn't even clearly show the Azores plume (only "somewhat resolved") and if, with some good will you see an Azores plume in a cross section (e.g. again with the Submachine website) it is actually also tilted, coming up from the south, similar to my models. Also, Nelson and Grand (Nature Geoscience, 2018) show a Yellowstone plume conduit that is tilted similarly to my model predictions. So, it is a misrepresentation if you write that my models "fail to explain the vertical shape of plumes in the lower mantle recently unraveled by tomographic models".

On a N-S cross-section, the Azores plume is not well-resolved (see Fig. 2 here-below). In order to account for dissenting interpretations of the tomographic model, we have deleted the corresponding remark Line 158-160: “These models fail to explain the southward elongated asymmetry across the Azores since they would predict an asymmetry in the opposite direction.”

Fig. 2: (a) E-W and (b) N-S seismic tomography cross-sections in the Mid-Atlantic, in the area of the Azores, with the model SEMUCB-WM1 of French and Romanowicz, 2015. Figures were plotted using the Submachine website (Hosseini et al, 2018, Shephard et al, 2017)

References have mistakes. I didn't check thoroughly, but here are a few I found

#2: Third author should be "Redfield"
 #4 and 31: Why "Science (80-.)"?
 #20 authors Solomon and Hung are missing
 (besides, I would rather write "Van der Voo" and "Van der Lee" as last names in #4 and #20)
 #27 write "McKenzie" with capital K.
 #40 M??ller needs to be fixed.
 Supplementary #6: "convection models" is missing.
 Besides, make it consistent which letters are capitalized.

Changed accordingly.

Figures in general: it would be useful and make the reading easier if there was also a scale, or at least tickmarks, on the right-hand side.

We added a clearer scale on Fig. 1. We also added a scalebar on the insets of Fig. 3a-d.

I find a reference viscosity of $1.03E22$ Pas, presumably used for the upper mantle, much too high. It is about 10 times higher than the classical "Haskell average" inferred from postglacial rebound, and much higher of other estimates of viscosity below the lithosphere. Also, if the lower mantle viscosity is 30 times higher, it would be about $3E23$ Pas, which I also consider much too high. I may interpret your numbers wrong, but if so, that would need clarification. With such (at least in my understanding) unrealistic values, I find it actually quite surprising that your results, at least in terms of plume size and spacing, look quite realistic, similar to, for example Hassan et al. (G-cubed, 2015) who use, in my view, a much more realistic viscosity structure.

We have now added the time-averaged geotherms and the corresponding time-averaged viscosity profiles (calculated directly from the geotherms) for our model on Fig. SI 9. As shown on Fig. SI 9 and the Fig. 3 here-below, the upper mantle mean viscosity (green curve) is about 2 orders of magnitude lower than the reference viscosity (10^{20} Pa.s) and the lower mantle mean viscosity (green curve) varies between 10^{22} and 10^{23} Pa.s.

This profile is rather similar to Hassan et al., 2015 profile (see Fig. 3 here-below, black dotted curve), which explains why we find similar plume size and spacing.

Fig. 3: Left: time-averaged minimum (blue curve), mean (green curve) and maximum (red curve) geotherm of our numerical model. **Right:** geotherm-derived minimum (blue curve), mean (green curve) and maximum viscosity profile of our numerical model. The transparent green curve corresponds to the time and depth-averaged viscosity profile of our model. In the latter, the presence of highly-viscous slabs in the mantle shifts the average towards larger viscosities compared to the green curve. The black dotted curve shows the viscosity profile used in Hassan et al., 2015.

What I find a bit hard to understand, though, is why your plumes have a volume flux and heat flux about a factor ~ 8 lower in the lower mantle than in the upper mantle. As I understand your equations, your computations are incompressible, so the plume volume flux should remain the same through the mantle, simply because of mass

conservation (what flows in in the bottom should come out in the top). Of course, there may be some entrainment, but increase by a factor ~ 8 appears quite unrealistic to me.

The code that we use (StagYY) was indeed benchmarked to ensure mass conservation. The apparent discrepancy between our lower mantle and upper mantle volume and heat fluxes comes from our detection method of plumes.

We detect plumes on a map with a connectivity algorithm based on Labrosse et al., 2002, applied on the radial heat advection flow. We worked on two maps: 350 km for the upper mantle and 1000 km for the lower mantle. For each map, we chose independently two different heat advection thresholds to detect plumes (Fig. 4a and 4c here-below). This results in different distributions of cross-sectional areas in the upper and the lower mantle (Fig. 4b here-below) and to an underestimate of the cross-sectional area of plumes in the lower mantle (and thus of the resulting volume and heat fluxes) compared to the upper-mantle. We carefully chose our threshold in the upper mantle since we needed a critical comparison with data, but we did it less carefully in the lower mantle and we apologize. Carefully means that we benchmark our detection performed with heat advection by comparing the result with residual temperature, velocity and viscosity maps and verify that detections are consistent in time and depth over the upper mantle.

Since we do not use the lower mantle histograms shown on Fig. 4 here-below are not critical for our main argumentation about the motion of the Azores plume, we chose not to add them in the new version of our manuscript (Fig SI 10), and save the time needed to benchmark the lower mantle detection.

Fig. 4 – Characteristics of all modelled plumes detected in the upper (350 km, blue) and the lower (1,000 km, orange) mantle throughout the model time integration. Plume (a) temperature excess, (b) cross-sectional area and equivalent radius (using the approximation of a circular conduit), (c) rising speed, (d) volume flux, (e) heat flux and (f) tilt angle between the lower (1,000 km) and the upper (350 km) mantle.

Reviewers' comments:

Reviewer #3 (Remarks to the Author):

I read the manuscript revision and the technical points have mostly been addressed, so I think the paper is basically acceptable with no major changes. I think the paper will be an important contribution towards understanding the interaction of plumes and ridges, combining geochemistry (which I cannot say much about) and geodynamic modelling (which I can assure is technically very well-done and at the forefront of the discipline). But on a more fundamental level I still have a different view -- that, in order to "show" something with a numerical model, you would really have to exclude all other possible causes. Otherwise you can rather just corroborate or provide support for a certain scenario. In the case you address here another possible explanation would be for example that the material sampled by the ridge the south is hotter because that segment of the ridge is more closely overlying the margin of the African Large Low Shear Velocity Province, or more specifically if there is a separate hotspot - the "New England hotspot", although it may be extinct by now. Not that I would favour that scenario, but I think it is possible, and there are probably other possible explanations which I can't even think of.

Also -- although the numerical model results show certain similarities with the actual situation -- there are still differences. In particular, in a reference frame where the African plate has been nearly stationary in that region, the North American plate would have overlain the hotspot some time ago and moved essentially westward with about 2 cm/yr (and I presume that not only holds for all global moving hotspot reference frames, i.e. not only Doubrovine et al., 2012 but also for Steinberger et al., 2004 and Torsvik et al., 2008 although I didn't re-compute now) and because the ridge extends from Azores towards the southwest, there is a component of plate motion in the direction of the ridge which could contribute to generation of the wake. Or there could be a combination of effects - northward hotspot motion combined with (on average) westward plate motion could combine to produce a wake that extends to the southwest. Also, because of the asymmetric plate motions, the ridge wasn't where it is now in the past, e.g. 50 Myr ago it was about 500 km further east. So I'm not sure whether in this situation this mechanism would produce a wake at the ridge, and not rather below the African plate where (in an absolute reference frame) the ridge was located in the past.

Apart from that I have few small comments left:

line 33: "narrow ... conduit": At least the global study by French and Romanowicz doesn't see "narrow" conduits. Because they even write in their title of "broad" plumes.

line 133: I don't know why you write here "considering St-Helena and Tristan hotspots as fixed" when four lines further down of "several reference frames based on moving hotspots". I think you can probably remove the phrase "In a reference considering St-Helena and Tristan hotspots as fixed" without losing any meaning.

line 285: Write "McKenzie" with capital K and there is an extra space before the comma.

In general (not only this reference) I find the capitalization in the reference titles inconsistent. I

think only proper names should be capitalized, but for some references more words are capitalized. Also, page or paper numbers are missing or wrong in a number of cases, e.g.

Steinberger et al. (2004) should be 167-173, Gente Maia and Goslin shouldn't have page numbers, but a paper number 8514 instead, Thorne and Garnero should be paper number B08301; for your own previous paper, the title is incomplete (in the main paper, not the supplement) and the page numbers should be 3140-3163.

line 422: This is now Fig. SI11 not SI10.

Bernhard Steinberger

One-by-one point response to reviewers

Reviewer #3 (Remarks to the Author):

I read the manuscript revision and the technical points have mostly been addressed, so I think the paper is basically acceptable with no major changes. I think the paper will be an important contribution towards understanding the interaction of plumes and ridges, combining geochemistry (which I cannot say much about) and geodynamic modelling (which I can assure is technically very well-done and at the forefront of the discipline).

But on a more fundamental level I still have a different view -- that, in order to "show" something with a numerical model, you would really have to exclude all other possible causes. Otherwise you can rather just corroborate or provide support for a certain scenario. In the case you address here, another possible explanation would be for example that the material sampled by the ridge the south is hotter because that segment of the ridge is more closely overlying the margin of the African Large Low Shear Velocity Province, or more specifically if there is a separate hotspot - the "New England hotspot", although it may be extinct by now. Not that I would favour that scenario, but I think it is possible, and there are probably other possible explanations which I can't even think of.

To temper our view, we have modified the sentences Line 23 : "Using for the first time a 3D spherical mantle convection where plumes, ridges and plates interact in a fully dynamic way, we explain how the extent, shape and asymmetry of this anomaly is a consequence of the Azores plume moving northwards by 1-2 cm/yr, independently from other Atlantic plumes, possibly developing the Great-Meteor and Corner Rise volcanic chains." and Line 58: "Accounting for the kinematic context around the Azores, our model reveals that a northward motion of the plume > 1 cm/yr explains the geometry of the observed asymmetry."

Fig. 1: Spatial distribution of samples as a function of their evolution through magma differentiation. (a) Blue dots correspond to whole-rock and glasses analyses (100% of the dataset). Red circles: hot spots from Courtillot et al (2003). (b) Distribution of (zero-age) MORBs giving a solution of calculation with PRIMELT3. Mantle potential temperatures (T_P) were calculated with reduced conditions ($Fe^{2+}/\Sigma Fe = 0.9$) in the source and filtered with $MgO < 8$ wt%. Here, 100% of the dataset have been reported.

We don't think that our results show an influence of the New England-Great Meteor hotspot interaction with the ridge, since Fig. 1a and Fig. 2 show a linear decrease of mantle potential temperature, of cpx fractionation and of chemical evolution (glasses or aphyric basalts) from the Azores to the south instead of a see-saw evolution that would be characteristic of the presence of another plume. Moreover, the last markers of the New England hotspot activity are 11 to 16 Ma at least (Wendt et al., 1976).

In the case of a potential implication of the African LLSVP in the thermal anomaly along the MAR south the Azores, we would expect to also detect from our petrological method such an anomaly in the South-Atlantic, where the Atlantic ridge also overlies a margin of the LLSVP. From Fig. SI 6 and Fig. 1 here-above, we do not detect such thermal anomalies in the vicinity of both the African and the Pacific LLSVPs with our method PRIMELT (although Dalton et al, 2014 suggested a possible influence of the Pacific LLSVP on the high temperature anomalies along the East-Pacific Rise). Instead, our maps show that we consistently detect anomalously hot mantle sections in the areas where hotspot-ridge interactions are the most likely.

Also -- although the numerical model results show certain similarities with the actual situation -- there are still differences. In particular, in a reference frame where the African plate has been nearly stationary in that region, the North American plate would have overlain the hotspot some time ago and moved essentially westward with about 2 cm/yr (and I presume that not only holds for all global moving hotspot reference frames, i.e. not only Doubrovine et al., 2012 but also for Steinberger et al., 2004 and Torsvik et al., 2008 although I didn't re-compute now) and because the ridge extends from Azores towards the southwest, there is a component of plate motion in the direction of the ridge which could contribute to generation of the wake. Or there could be a combination of effects - northward hotspot motion combined with (on average) westward plate motion could combine to produce a wake that extends to the southwest.

We think that the south-westward orientation of the MAR axis south the Azores cannot explain, by itself, the observed geometry of the thermal anomaly of the plume. Indeed, as shown on Fig. 1a, in the case of a symmetric wake, the section of the MAR north the Azores would sample the plume wake over a larger area than it is actually. We added a mention to the geometry of the MAR axis Line 136-138: "The south-westward orientation of the MAR south the Azores cannot explain by itself the much larger extent of the thermal anomaly of the plume to the south (Fig. 1a)."

We have rewritten the paragraph Line 145-150 to make clearer the potential combination of a northward motion of the Azores plume and a westward drift of North-America on the asymmetry of the thermal anomaly along the MAR: "Using a global moving hotspot reference frame, Torsvik et al.³⁹, Steinberger et al.³⁸ and Doubrovine et al.⁴⁰ proposed stable African and European plates and suggested a westward-drifting North-America by 2 cm/yr (corresponding to case 3, Fig. 3c). Accounting for a stationary or eastward-moving plume would generate a westward-oriented asymmetric wake which would result in a symmetric thermal anomaly along the MAR axis (Fig. 3e). Therefore, only a combination between a northward motion of the Azores plume and such plate motions would explain the observed southward asymmetry of its wake."

Also, because of the asymmetric plate motions, the ridge wasn't where it is now in the past, e.g. 50 Myr ago it was about 500 km further east. So, I'm not sure whether in this situation this mechanism would produce a wake at the ridge, and not rather below the African plate where (in an absolute reference frame) the ridge was located in the past.

On Fig. 2 here-below (corresponding to Fig. SI 13), we show the interaction of a moving plume with a ridge axis comprising one segment moving westward (ridge axis 2). Although the plume departs from the ridge from 60 Myr, we still observe an asymmetric temperature profile along ridge axis 2 at 100 Myr (yellow line on Fig. 1c here-below). Therefore, while the MAR has moved westward over the last 50 Ma, and in the case the Azores plume had moved northward parallel to this moving ridge axis during this time period, we would still be able to observe an asymmetric signal along the ridge nowadays. Moreover, in the case the Azores plume has participated to the building of Corner Rise and Great Meteor seamounts from 80 Ma, the reconstructed location of these seamounts suggests that melting of plume material occurred along the ridge axis. We have included the following discussion about the migration of the Atlantic ridge over the past 50 Myr, Line 154-158: "Finally, the asymmetric motions of Africa and North-America predict a westward migration of the MAR axis during the last 50 Ma at a speed of about 1.5 cm/yr⁶. Although this would possibly result in a shift between the location of the center of the wake of the Azores plume and the MAR, our model shows that the potential temperature along a ridge having interacted with a plume at least 40 Myr before still records the asymmetry related to the relative motion between the ridge and the plume (Fig. SI 13)."

Fig. 2: Interaction of a moving plume with a moving ridge. (a) Temporal evolution of the relative position of a ridge (straight colored lines) and a mantle plume in our global model of mantle convection with plate-like behavior. Colored contour lines outline the 1480°C isotherm at 10 Myr (magenta), 40 Myr (red), 70 Myr (orange) and 100 Myr (yellow). (b-c) Temporal evolution of the potential temperature distribution at 110 km depth along axis 1 (b) and 2 (c). The colored arrows correspond to the location of the plume maximum of temperature at corresponding timesteps. Note the westward drifting ridge axis 2 and the development of an asymmetric potential temperature profile from 70 Myr along ridge axis 2.

Apart from that I have few small comments left:

line 33: "narrow ... conduit": At least the global study by French and Romanowicz doesn't see "narrow" conduits. Because they even write in their title of "broad" plumes.

Changed accordingly.

line 133: I don't know why you write here "considering St-Helena and Tristan hotspots as fixed" when four lines further down of "several reference frames based on moving hotspots". I think you can probably remove the phrase "In a reference considering St-Helena and Tristan hotspots as fixed" without losing any meaning.

Changed accordingly.

line 285: Write "McKenzie" with capital K and there is an extra space before the comma. In general (not only this reference) I find the capitalization in the reference titles inconsistent. I think only proper names should be capitalized, but for some references more words are capitalized. Also, page or paper numbers are missing or wrong in a number of cases, e.g. Steinberger et al. (2004) should be 167-173, Gente Maia and Goslin shouldn't have page numbers, but a paper number 8514 instead, Thorne and Garnero should be paper number B08301; for your own previous paper, the title is incomplete (in the main paper, not the supplement) and the page numbers should be 3140-3163.

line 422: This is now Fig. S111 not S110.

Changed accordingly.

Bernhard Steinberger